EMBO
Molecular Medicine

# Allele-specific silencing therapy for Dynamin 2-related dominant centronuclear myopathy

Delphine Trochet[1] [ID], Bernard Prudhon[1], Maud Beuvin[1], Cécile Peccate[1], Stéphanie Lorain[1], Laura Julien[1], Sofia Benkhelifa-Ziyyat[1], Aymen Rabai[2], Kamel Mamchaoui[1], Arnaud Ferry[1], Jocelyn Laporte[2], Pascale Guicheney[3] [ID], Stéphane Vassilopoulos[1] & Marc Bitoun[1],[*] [ID]

## Abstract

**Rapid advances in allele-specific silencing by RNA interference established a strategy of choice to cure dominant inherited diseases by targeting mutant alleles. We used this strategy for autosomal-dominant centronuclear myopathy (CNM), a rare neuromuscular disorder without available treatment due to heterozygous mutations in the *DNM2* gene encoding Dynamin 2. Allele-specific siRNA sequences were developed in order to specifically knock down the human and murine *DNM2*-mRNA harbouring the p.R465W mutation without affecting the wild-type allele. Functional restoration was achieved in muscle from a knock-in mouse model and in patient-derived fibroblasts, both expressing the most frequently encountered mutation in patients. Restoring either muscle force in a CNM mouse model or DNM2 function in patient-derived cells is an essential breakthrough towards future gene-based therapy for dominant centronuclear myopathy.**

**Keywords** allele-specific silencing therapy; centronuclear myopathy; Dynamin 2; RNA interference

**Subject Categories** Genetics, Gene Therapy & Genetic Disease; Musculoskeletal System

## Introduction

Autosomal-dominant centronuclear myopathy (AD-CNM) is a rare congenital myopathy without available curative treatment and with pathomechanisms still largely unknown. Autosomal-dominant centronuclear myopathy exhibits a wide clinical spectrum from severe-neonatal to mild-adult forms. The most frequent phenotype corresponds to late-childhood or adult onset forms in which motor milestones are delayed and diffuse skeletal muscle weakness mainly involves facial and limb muscles (Fischer *et al*, 2006; Hanisch *et al*, 2011). Muscle weakness is slowly progressive but loss of independent ambulation may occur during the fifth decade. In the vast majority of patients, respiratory and cardiac functions are normal. In the severe and early-onset CNM (Bitoun *et al*, 2007), paediatric patients usually have generalized weakness, hypotonia, facial weakness with open mouth, ptosis and ophthalmoplegia. Progression may be fatal (Jungbluth *et al*, 2010) but is most often slowly progressive (Melberg *et al*, 2010; Susman *et al*, 2010). Few children can improve their strength during early childhood (Susman *et al*, 2010), but some of them can still develop a restrictive respiratory syndrome at later ages (Bitoun *et al*, 2007; Melberg *et al*, 2010). The histological features in muscle biopsies consist of nuclear centralization associated with atrophy and predominance of type 1 fibres and radial arrangement of sarcoplasmic strands radiating from the central nuclei.

Autosomal-dominant centronuclear myopathy results from mutations in the *DNM2* gene which encodes Dynamin 2 (DNM2) (Bitoun *et al*, 2005) also involved in rare cases of Charcot–Marie–Tooth disease (Zuchner *et al*, 2005) and hereditary spastic paraplegia (Sambuughin *et al*, 2015). DNM2 belongs to the superfamily of large GTPases (Heymann & Hinshaw, 2009) and acts as a mechanochemical scaffolding molecule that can deform biological membranes leading to the release of vesicles from distinct membrane compartments. At the plasma membrane, DNM2 is involved in clathrin-dependent and clathrin-independent endocytosis. DNM2 is also involved in the formation of vesicles from endosomes and trans-Golgi network. Furthermore, several studies have highlighted the role of DNM2 as a regulator of actin and microtubule cytoskeleton networks (Durieux *et al*, 2010a).

The DNM2 protein is ubiquitously expressed, and to date, there is no explanation for the tissue-specific impact of the *DNM2* mutations. Several DNM2-dependent processes have been shown to be impaired by CNM mutations and supposed to contribute to muscle pathophysiological mechanisms (i.e. endocytosis, microtubules network and recently actin-mediated trafficking) (Durieux *et al*, 2010a; Gonzalez-Jamett *et al*, 2017). In mouse muscle fibre, DNM2 presents a striated transversal staining pattern on the I-band of the sarcomere centred on the Z-line. DNM2 localized to the perinuclear MTOC, Golgi apparatus, microtubules, sarcoplasmic reticulum, is

1 Research Center for Myology, UPMC Univ Paris 06 and INSERM UMRS 974, Institute of Myology, Sorbonne Universités, Paris, France
2 Department of Translational Medicine and Neurogenetics, IGBMC, INSERM U964, CNRS UMR7104, Collège de France, University of Strasbourg, Illkirch, France
3 Institute of Cardiometabolism and Nutrition (ICAN), INSERM UMR_S1166, UPMC Univ Paris 06, Sorbonne Universités, Paris, France
*Corresponding author. Tel: +33 1 42 16 57 18; E-mail: m.bitoun@institut-myologie.org

enriched at the neuromuscular junction and colocalized with clathrin heavy chain (Durieux *et al*, 2010b). *DNM2* mutations in AD-CNM patients are mostly missense (Bohm *et al*, 2012), and when tested, the mutant protein is expressed normally (Bitoun *et al*, 2005, 2009). Mutations are thought to be responsible for a gain of function and/or a dominant negative effect through an increased GTPase activity and formation of abnormal stable Dnm2 oligomers (Kenniston & Lemmon, 2010; Wang *et al*, 2010). We have previously developed a knock-in (KI) mouse model expressing the most frequent *DNM2*-CNM mutation (found in 30% of patients), that is the KI-*Dnm2*[R465W] model (Durieux *et al*, 2010b). Heterozygous KI-*Dnm2* mice progressively develop features of the human CNM including impairment of force generation, muscle atrophy and altered spatial organization of the muscle fibre's oxidative compartments.

Whereas the full *Dnm2* knock-out is lethal at embryonic stages in mice, heterozygous knock-out mice expressing 50% of *Dnm2* are viable with unaffected muscle function (Ferguson *et al*, 2009; Tinelli *et al*, 2013; Cowling *et al*, 2014). A potential therapeutic approach is therefore the suppression of the mutant allele expression without reducing the wild-type allele. With this objective, allele-specific RNA interference (AS-RNAi) emerged as a powerful strategy for dominant inherited diseases (Trochet *et al*, 2015). Therapeutic benefit has been demonstrated in cells from patients (Klootwijk *et al*, 2008; Sierant *et al*, 2011; Kaplan *et al*, 2012; Muller *et al*, 2012), animal models (Xia *et al*, 2006; Jiang *et al*, 2013) and in a first clinical trial for an inherited skin disorder (Leachman *et al*, 2010). Here, we report the proof of principle of AS-RNAi therapy for AD-CNM. By applying this strategy, functional rescue was achieved in the KI-*Dnm2*[R465W] mouse model and in patient-derived

fibroblasts, both expressing the most frequently encountered mutation in patients.

## Results

### Identification of allele-specific siRNAs in heterozygous cells

In the KI-*Dnm2*[R465W] mouse model, the missense mutation corresponds to a single-point mutation in exon 11 (c.1393 A>T, p.R465W). A screening for allele-specific siRNAs capable of silencing the mutant *Dnm2* allele without affecting the WT allele was performed in mouse embryonic fibroblasts (MEFs) cultured from heterozygous (HTZ) KI-*Dnm2* embryos. We developed RT–PCR assay to discriminate the WT and mutant alleles after restriction enzyme digestion performed at the end of the exponential phase of PCR amplification (Appendix Fig S1). Using this assay, we assessed allele-specific properties of 12 siRNAs among the 19 possible siRNAs (Fig 1A). At low concentration (20 nM) of scramble siRNA, HTZ MEFs showed a mutant/WT ratio equal to 1 in agreement with similar expression of both WT and mutant alleles (Fig 1B). Among the 12 assessed siRNAs, six siRNAs (si9, si10, si11, si12, si15 and si16) exhibited allele-specific silencing properties as demonstrated by significant reduction in mutant/WT ratios compared to scramble siRNA-transfected cells (Fig 1B).

Further analyses were pursued for two of the most efficient siRNA sequences, that is si9 and si10 at higher concentrations (100 nM). After 48 h, semi-quantitative RT–PCR showed around 50% reduction in total *Dnm2*-mRNA expression (WT + mutant) for each siRNA (Fig 2A). Amplicon sequencing showed the HTZ *Dnm2*

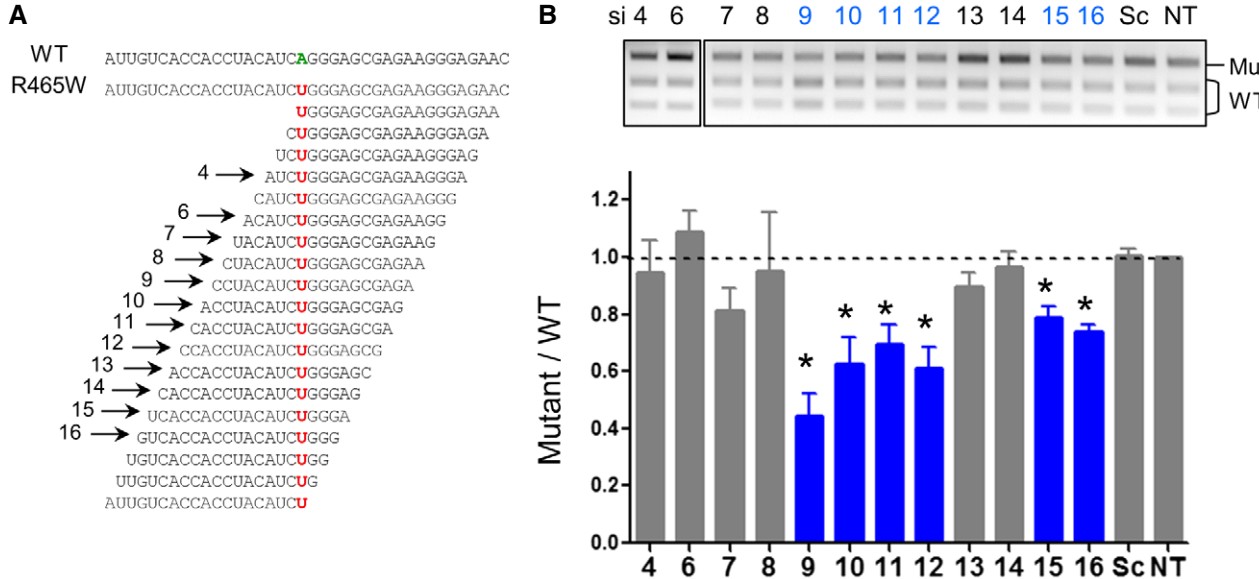

**Figure 1.  Identification of six allele-specific siRNAs in MEFs.**

A   Wild-type (WT) and mutant (R465W) mRNA sequences in the region of the mutation. The sequences of the 19 possible siRNAs targeting the mutation (in red) are indicated. Arrows show the sense strand of the 12 assessed siRNAs numbered relative to the position of the mismatch between siRNA and WT sequences.

B   EcoNI digestion profile of the *Dnm2* RT–PCR products. Histogram represents mean ± SEM of calculated mutant/WT ratio for siRNAs transfected at 20 nM for 48 h. *$P < 0.05$, two-tailed using a Mann–Whitney $U$-test compared to scramble ($n = 4$).

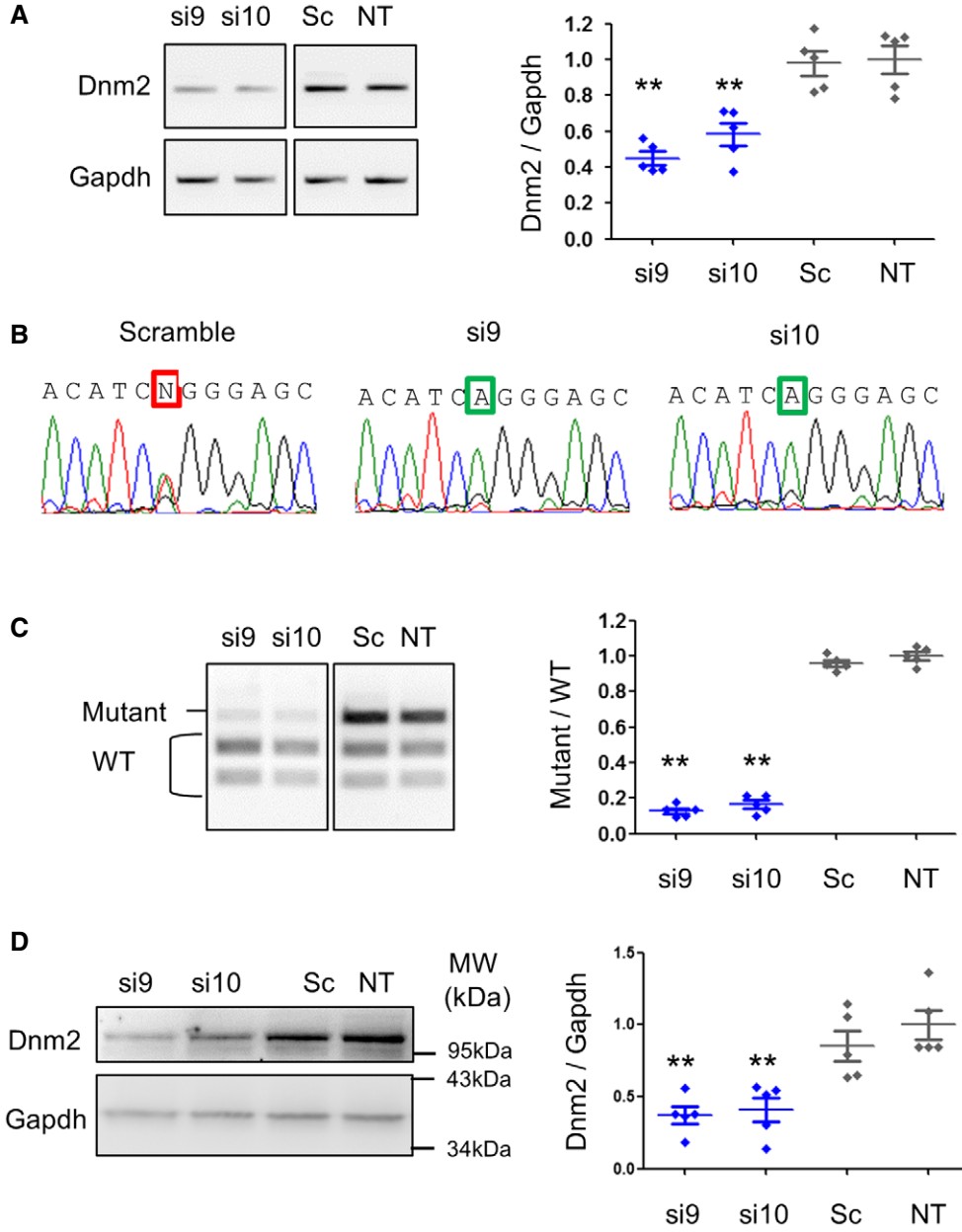

**Figure 2. Si9 and si10 are potent allele-specific siRNAs in MEFs.**

A    Semi-quantitative *Dnm2* and *Gapdh* RT–PCR products and quantification of *Dnm2* expression normalized to *Gapdh*.
B    Sequence of *Dnm2* amplicons from cells transfected with si9, si10 and scramble siRNAs. Squares indicate the mutant nucleotide (N = T and A).
C    EcoNI digestion profile *Dnm2* PCR and quantification of the mutant/WT ratio.
D    Dnm2 Western blot and quantification of signal by densitometry. Gapdh was used as loading control.

Data information: In scatter plots (A, C and D), the bars are mean values and error bars indicate SEM. **$P < 0.01$, one-tailed using a Mann–Whitney *U*-test compared to scramble ($n = 5$). In (A–D), the siRNAs were transfected at 100 nM for 48 h. Sc: scramble siRNA. NT: non-transfected.

sequence in scramble-transfected cells but only the WT sequence in si9- and si10-transfected cells (Fig 2B). Allele specificity of the two siRNAs against the mutant allele was demonstrated by the mutant/WT ratio reduction (Fig 2C) and confirmed by quantification of mutant and WT mRNA expression relative to the housekeeping Gapdh (Appendix Fig S2A). Both si9 and si10 also reduced Dnm2 protein content around 50% as established by Western blot (Fig 2D)

without modifying Dnm2 subcellular localization (Appendix Fig S2B). Under these conditions, expression of Dnm1 and Dnm3 transcripts which exhibit 4 and 5 mismatches with the two siRNAs, respectively, was not modified by si9 and si10 (Appendix Fig S2C– F). Altogether, these data validate si9 and si10 as efficient allele-specific siRNA sequences as both specifically knock down the mutant *Dnm2* transcript and reduce *Dnm2* transcript and protein

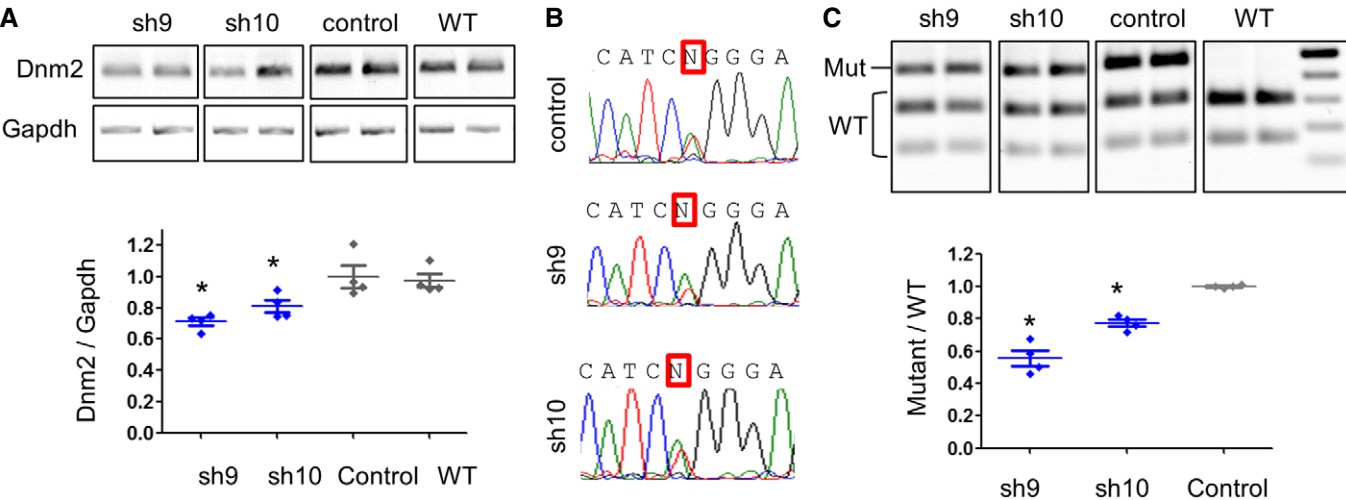

**Figure 3. Sh9 is efficient to specifically reduce the mutant allele in 3-month-treated muscle from young mice.**

A   *Dnm2* and *Gapdh* RT–PCR products from AAV-shRNA-transduced muscles and quantification of *Dnm2* expression normalized to *Gapdh*. WT muscles were included as control.

B   Sequence of *Dnm2* amplicons from transduced TA muscles. Squares indicate the mutant nucleotide (N = T and A).

C   EcoNI digestion profile of the *Dnm2* amplicons and quantification of the mutant/WT ratio.

Data information: In scatter plots (A, C), the bars are mean values and error bars indicate SEM. *$P < 0.05$, one-tailed using a Mann–Whitney $U$-test compared to AAV-control ($n = 4$).

levels by half in HTZ MEFs. The ability of 100 nM of si9 and si10 to specifically silence the mutant allele was confirmed by RT–PCR in immortalized mouse myoblasts derived from heterozygous (HTZ) KI-*Dnm2* (Appendix Fig S3).

**Early treatment of muscle phenotype in KI-*Dnm2* mice**

At 3 weeks of age, HTZ mice exhibit a robust decrease in contractile properties in Tibialis anterior (TA) muscle whereas muscle mass and histology are still normal compared to WT animals. After 2 months of age, contractile properties, muscle mass and histology are affected (Durieux *et al*, 2010b). Adeno-associated virus (AAV) serotype 1 vectors containing small hairpin (sh) RNA corresponding to si9 and si10 (AAV-sh9 and AAV-sh10, respectively) and a control AAV without shRNA sequence (AAV-control) were designed for *in vivo* evaluation. We first assessed efficacy of early treatment relative to the disease's time-course. AAVs were injected intramuscularly at $10^{11}$ viral genomes (vg)/TA muscle of HTZ KI-*Dnm2* mice at 1 month of age, and muscle phenotype was investigated 1 and 3 months later (groups 1M-1M and 1M-3M, respectively).

At the end point of the 3-month treatment period, expression of *Dnm2* transcript was quantified by RT–PCR showing a significant decrease in the *Dnm2* content in sh9- and sh10-expressing muscle compared to AAV-control values (−30 and −20%, respectively) (Fig 3A). Sequencing *Dnm2* amplicons showed a HTZ sequence at the mutant nucleotide position in AAV-transduced muscles with a reduction in the peak corresponding to the mutant nucleotide with sh9 and to a lesser extent with sh10 (Fig 3B). Allele-specific silencing was demonstrated by reduced mutant/WT ratio using RT–PCR and EcoNI digestion profile (Fig 3C). Specific silencing of the mutant transcript was also shown by the quantification of the digested products corresponding to mutant and WT mRNAs relative

to Gapdh expression (Appendix Fig S4A) and for sh9 by a second RT–PCR assay using primers designed for specific amplification of either WT or mutant alleles (Appendix Fig S4B and C). These data confirmed that sh9, and to a lesser extent sh10, maintained allele-specific silencing properties *in vivo*.

We evaluated capacity of 3-month sh9 and sh10 treatments to alleviate muscle atrophy, morphological abnormalities and contractile properties. When compared to WT muscle, HTZ TA muscles showed a significant 28% decrease in mass (Fig 4A). In AAV-sh9-injected mice, muscle mass was restored close to WT values while AAV-sh10 treatment led to partial intermediate improvement of muscle atrophy (Fig 4A). Quantification of fibre diameter frequency (Fig 4B) showed total rescue of fibre size only in muscles transduced with AAV-sh9. Under these conditions, the disease-specific histopathological abnormalities, that is central accumulation of oxidative cell compartments on DPNH oxidative staining, were almost absent in sh9-expressing muscles but were still present with sh10 (Fig 4C and D). Decrease in absolute maximal force (−40%) and in specific maximal force (−15%), present in HTZ muscles compared to WT, was almost circumvented by sh9 expression (Fig 4E and F). As noticed for the muscle mass, functional benefit was incomplete for sh10.

Results were more moderate for HTZ KI-*Dnm2* mice injected with AAV-sh9 and 10 for a shorter period (1-month treatment) (Appendix Fig S5). Changes in the total *Dnm2* transcript levels were undetectable, and the mutant transcript was only slightly reduced in treated mice (Appendix Fig S5A and B). Functionally, partial but significant improvements of the absolute and specific maximal forces are observed at this stage only with the sh9 treatment (Appendix Fig S5C and D). However, none of the treatment is sufficient after 1 month to alleviate muscle atrophy and morphological abnormalities (Appendix Fig S5E–G). Altogether, these data

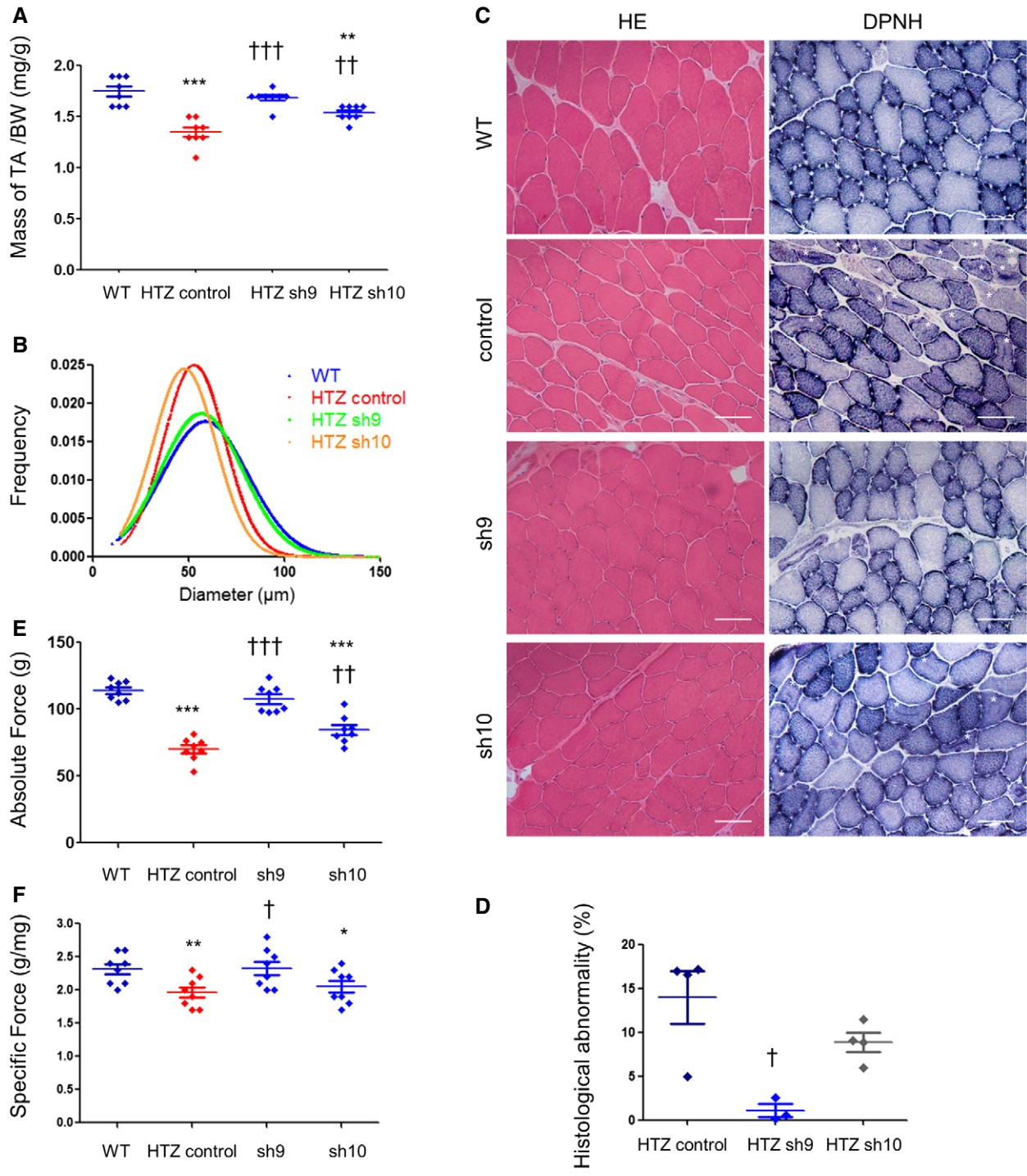

**Figure 4. Sh9 3-month treatment abolishes muscle defects in young mice.**

A    Muscle mass in AAV-shRNA-injected mice (n = 8). The TA weights were normalized by the total body weight (mg/g). BW: body weight.

B    Frequency of fibre size in TA muscles (n = 3 for WT and 4 for the AAV-injected HTZ muscles).

C    Histochemical staining of TA sections from WT and AAV-injected HTZ mice. HE: haematoxylin–eosin staining. DPNH: reduced diphosphopyridine nucleotide diaphorase staining. Asterisks indicate fibres with abnormal central accumulations. Scale bars = 50 μm.

D    Quantification of histological abnormality. Scatter plot represents individual percentages of histological abnormality from heterozygous control or treated mice (n ≥ 3).

E, F    Absolute maximal force (E) and specific maximal force (F) developed by TA muscles (n = 8).

Data information: In scatters blot (A, D–F), bars represent mean ± SEM. Statistical analysis was performed using a one-tailed Mann–Whitney *U*-test. *$P < 0.05$, **$P < 0.01$ and ***$P < 0.001$ compared to WT. †$P < 0.05$, ††$P < 0.01$ and †††$P < 0.001$ compared to AAV-control.

demonstrate the efficiency of 3-month treatment with the sh9 sequence to abolish the phenotype in young CNM mice.

### Late KI-*Dnm2* mice muscle phenotype treatment

We next evaluated therapeutic properties of sh9 in older mice after a similar 3-month treatment started at 6 months of age when muscle phenotype consists of impaired force, muscle atrophy and morphological abnormalities (Group 6M-3M) (Durieux *et al*, 2010b). We did not show significant variation in the total *Dnm2* transcript levels in older AAV-sh9-treated mice (Fig 5A), and the mutant transcript was only slightly reduced (Fig 5B). In agreement, morphological abnormalities were still present (Fig 5C and D) and muscle mass was unchanged (Fig 5E) in these mice. A significant improvement of contractile properties occurred in older mice too but still far from the WT values (Fig 5F and G). Overall, our data showed that a late treatment was less efficient to restore a healthy phenotype compared to a similar treatment in young animals.

### Interferon response and AAV transduction efficiency in early and late treatments

In order to identify mechanisms underlying different impact of sh9 in young and older mice, we first looked for potential induction of interferon response in mice treated for 3 months. Expression of two interferon-induced genes (*Oas1* and *Stat1*) was determined by RT–PCR. Similar expression of both genes was observed in non-injected WT muscles and in all HTZ mice irrespectively of whether the AAVs were injected in young and old mice (Fig 6A). This result along with absence of fibrosis and signs of muscle necrosis–regeneration in routine haematoxylin and eosin staining (Figs 4C and 5C) indicated that the weaker functional benefit in older mice was not due to a higher muscle toxicity. Next, we evaluated transduction efficiency through quantification of the amount of viral genomes per nanogram of DNA. At 1 month of age, a similar transduction efficacy was reached 3 months later in muscles injected with AAV-control, AAV-sh10 and AAV-sh9, ranging from 800 to 4,000 vg/ng (Fig 6B) and comparable results were obtained in the 1M-1M group (Appendix Fig S6). In contrast, values dropped to 90 vg/ng for AAV-control and 200 for AAV-sh9 when AAV was injected at 6 months of age. In order to confirm these results and to see the AAV distribution in muscle tissue sections at these different ages, we have injected the same dose of an AAV serotype 1 encoding the murine secreted embryonic alkaline phosphatase (muSEAP) reporter gene in the tibialis anterior of heterozygous mice at 1 and 6 months old. One month after the injection, the number of viral particles detected in tibialis anterior was around 10-fold higher in muscle injected at 1 month old than in muscle injected at 6 months old (12,600 vg/ng vs. 1,370 vg/ng, $n \geq 6$; Fig 6C). Consistently, the phosphatase activity is detected in all muscle fibres at 1 month while muSEAP was not detected in large area of muscle section in tibialis injected at 6 months old (Fig 6D). In the same time, the effect of a 10-fold increase in the virus titre on the transduction in old mice was evaluated. The transduction efficiency was greatly improved in 6-month-old mice by increasing the viral dose, since the number of vg/ng (8,300 vg/ng) was comparable to the young mice and the large majority of fibres expressed the phosphatase (Fig 6C and D).

Overall, our data demonstrate that moderate efficacy of treatment, when started at 6 months of age, was linked to a weaker capability to transduce muscle with AAV vectors at this age.

### Allele-specific silencing in patient-derived cells

The target sequence for si9 and si10 shares 84% identity (16 out of 19 base pairs) between mouse and human sequences (Appendix Fig S7A). Corresponding human-specific si9 and si10 were synthetized and evaluated in one patient-derived fibroblast cell line expressing the c.1393 C>T, p.R465W mutation. *DNM2* mRNA content was reduced around 50% in cells transfected with si9 at 50 nM after 48 h, whereas si10 appears ineffective (Fig 7A). Amplicon sequencing confirmed disappearance of the mutant mRNA in si9-transfected fibroblasts, while si10-transfected cells still expressed a mix of WT and mutant *DNM2* (Fig 7B). We developed a semi-quantitative RT–PCR assay in order to discriminate WT and mutant alleles after restriction enzyme digestion of the amplicon (Appendix Fig S7B and C). Using this assay, mutant/WT ratio was equal to 1 for non-transfected and scramble-transfected cells and significantly reduced to 0.8 by si10 and 0.2 by si9 (Fig 7C). After transfection of si9 and si10 for 48 h, the reduction in the DNM2 protein contents was demonstrated by Western blot (Fig 7D). Given that si9 exhibited all the expected properties of allele-specific siRNA in human cells, si9 was further investigated at higher dose (100 nM). At this dose, expression of *DNM2* transcript was still reduced around 50% (Appendix Fig S8A) and allele specificity against the mutant *DNM2* mRNA was maintained (Appendix Fig S8B) without observation of cell toxicity (Appendix Fig 7C). Basic Local Alignment Search Tool (BLAST) used to identify potential off-targets for si9 (sense and anti-sense strands) showed SLC9A8 among the nearest mRNA sequences with 68% identity corresponding to complete identity on 13 consecutive nucleotides (Appendix Fig S9A). We checked for potential si9-induced silencing of SLC9A8 mRNA by RT–PCR 48 h after transfection of siRNA at 100 nM. Compared to scramble siRNA, si9 did not affect expression of the SLC9A8 transcript (Fig 7E).

We further pursued evaluation of si9 properties by investigating functional rescue in patient-derived cells. Since clathrin-mediated endocytosis is impaired in CNM patient fibroblasts (Bitoun *et al*, 2009), we used a fluorescent transferrin uptake assay to evaluate the capability of si9 to restore normal endocytosis. Compared to two healthy control cell lines, 15-min transferrin uptake was reduced in scramble-transfected cells from the CNM patient but achieved normal value in cells transfected with si9 for 48 h before assay (Fig 7F). Transfection of si9 in control cell lines did not impact clathrin-mediated endocytosis.

## Discussion

During the past decade, AS-RNAi has emerged as a powerful therapeutic strategy for dominant inherited diseases (Trochet *et al*, 2015). This strategy benefits from outstanding specificity of siRNAs capable to discriminate two sequences, even when differing by only a single nucleotide. This property qualified allele-specific RNAi to target single nucleotide substitutions representing the majority of the 26 *DNM2* mutations identified in AD-CNM, especially for the

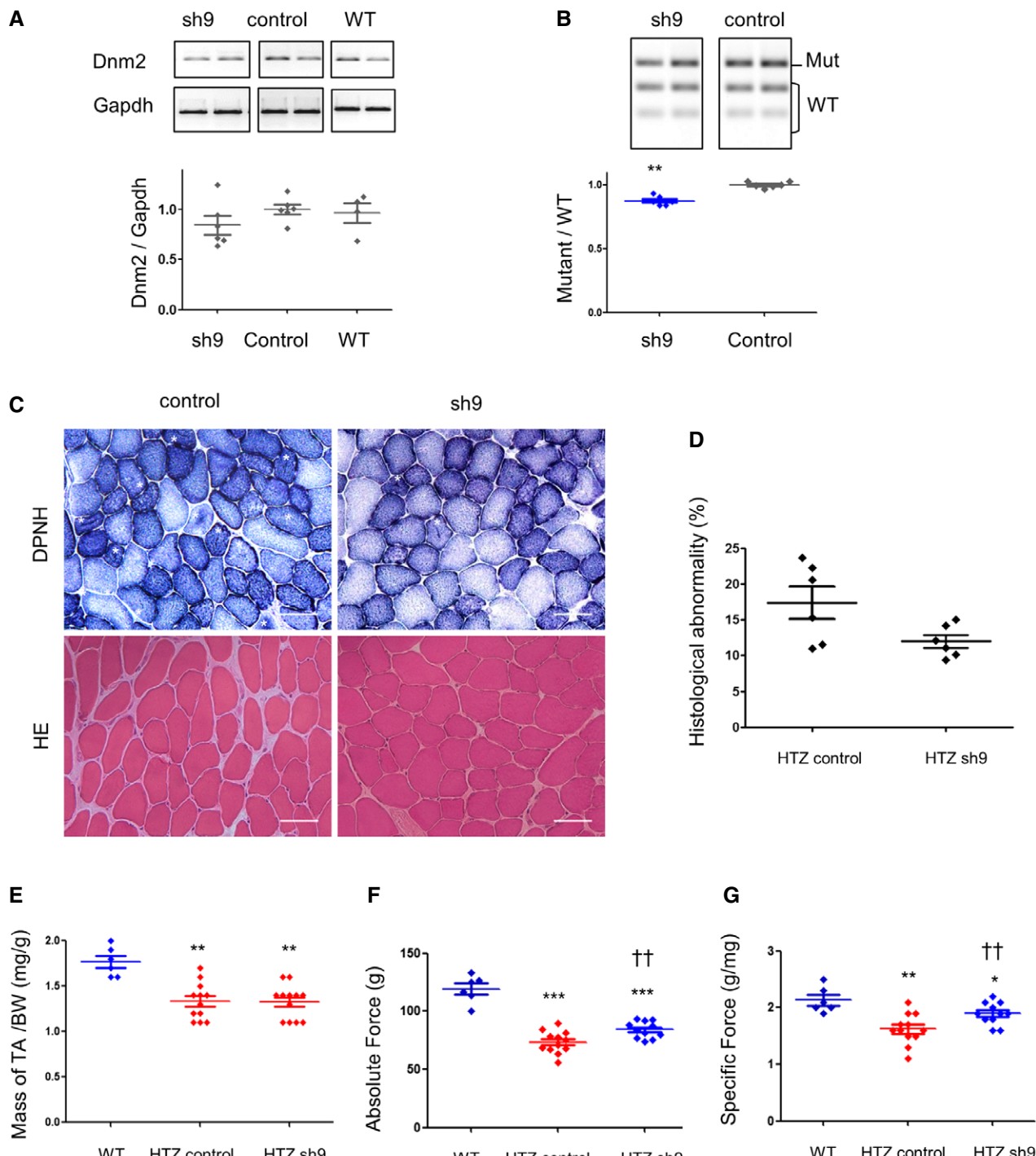

**Figure 5.  Sh9 is less efficient in old mice.**

A    *Dnm2* and *Gapdh* RT–PCR products from TA muscles and quantification of *Dnm2* expression normalized to *Gapdh*. WT muscles were included as control.

B    EcoNI digestion profile of the *Dnm2* amplicons and quantification of the mutant/WT ratio ($n = 6$).

C    Histochemical staining of TA sections from HTZ mice injected with AAV-sh9 or AAV-control. DPNH: reduced diphosphopyridine nucleotide diaphorase staining. Asterisks indicate fibres with abnormal central accumulations. Scale bars = 50 μm.

D    Quantification of histological abnormality. Scatter plot represents individual percentages of histological abnormality from heterozygous control or treated mice ($n = 6$).

E    Muscle mass normalized by the total body weight (mg/g) in AAV-injected mice ($n = 6$ for WT and $n = 12$ for HTZ).

F, G  Absolute maximal force (F) and specific maximal force (G) developed by TA muscles ($n = 6$ for WT and $n \geq 11$ for HTZ).

Data information: In (A, B, D–G), the bars in scatter plots represent the mean ± SEM. Statistical analysis was performed using a one-tailed Mann–Whitney *U*-test. *$P < 0.05$, **$P < 0.01$ and ***$P < 0.001$ compared to WT. ††$P < 0.001$ compared to HTZ-control.

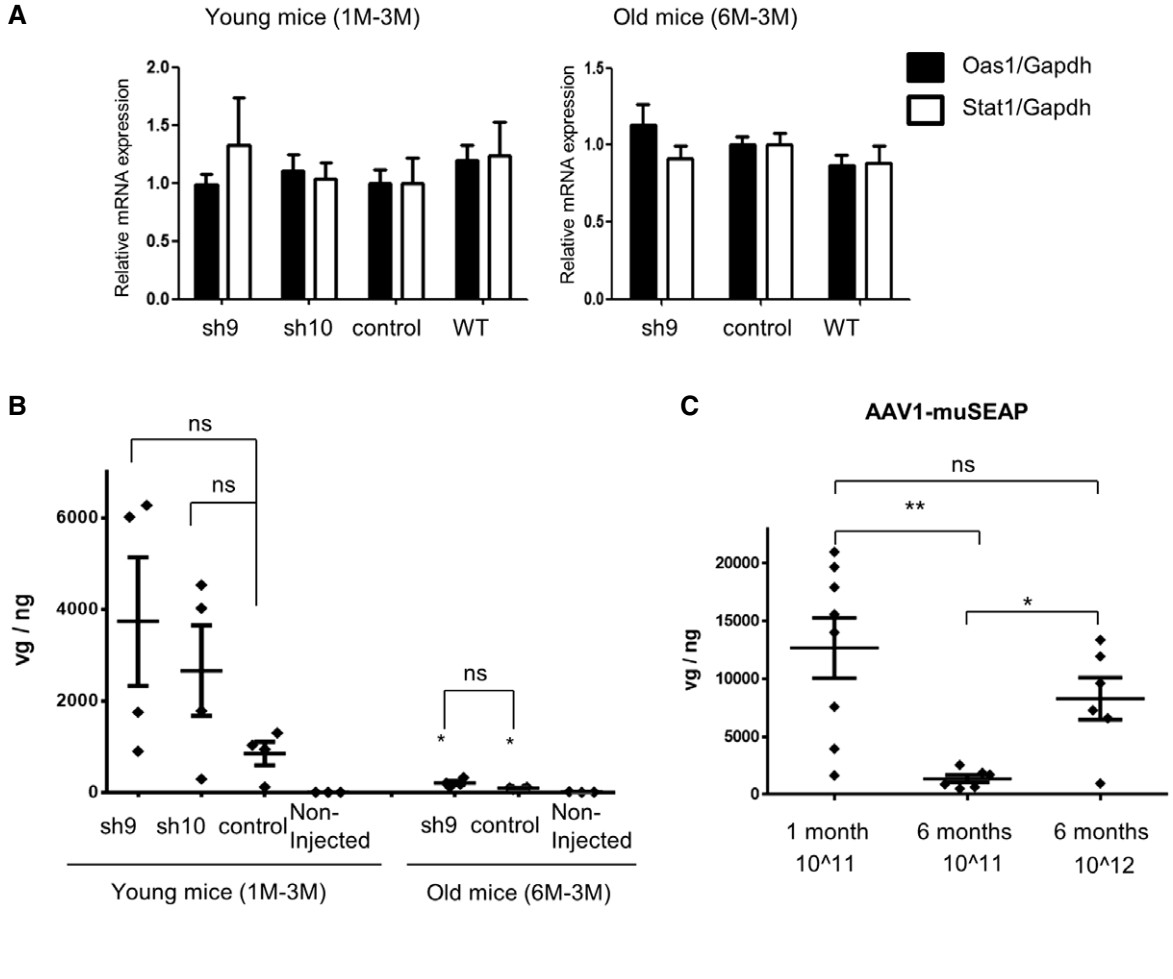

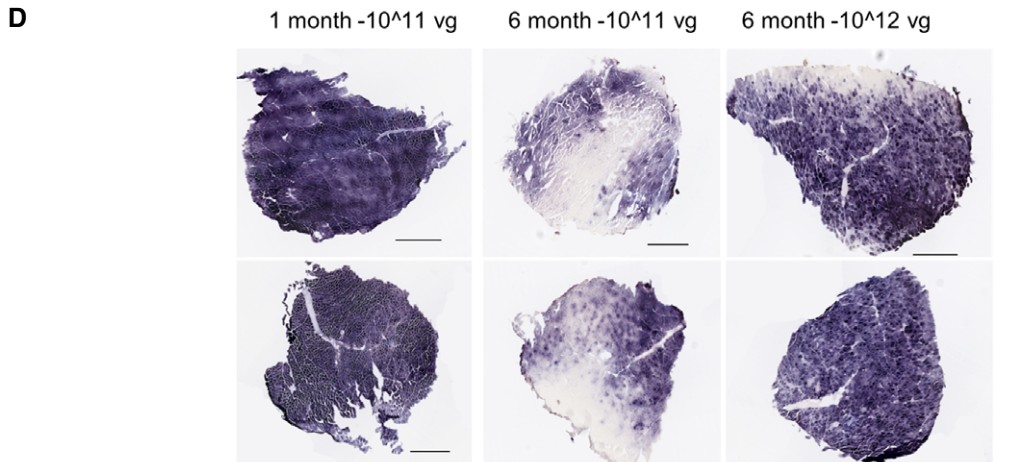

**Figure 6. Expression of interferon-induced genes and AAV transduction efficiency in young and old mice.**

A   Quantification of expression of the interferon-induced genes *Oas1* and *Stat1* mRNA relative to *Gapdh* mRNA by RT–PCR in muscle from mice treated for 3 months at 1 (young) and 6 (old) months of age. Left panel: young mice. Right panel: old mice. Histograms represent mean ± SEM. Statistical analysis was performed using a two-tailed Mann–Whitney *U*-test compared to non-injected WT values (young mice *n* = 4, old mice HTZ *n* = 6, WT *n* = 3).

B   Quantification of the viral genomes (vg) in mice injected with AAV-sh or control (*n* = 4). Non-injected WT muscles were included as negative control. *$P < 0.05$, two-tailed vs similar treatment at 1 month of age using a Mann–Whitney *U*-test. ns: non-significant.

C   Quantification of the viral genomes (vg) in mice injected with AAV-muSEAP (*n* ≥ 6). *$P < 0.05$, **$P < 0.01$, two-tailed Mann–Whitney *U*-test. ns: non-significant.

D   Histochemical detection of muSEAP performed in mice tibialis anterior (TA) section one month after the injection. Representative staining is shown. Scale bars = 500 μm.

Data information: Scatter plot bars represent mean ± SEM.

    

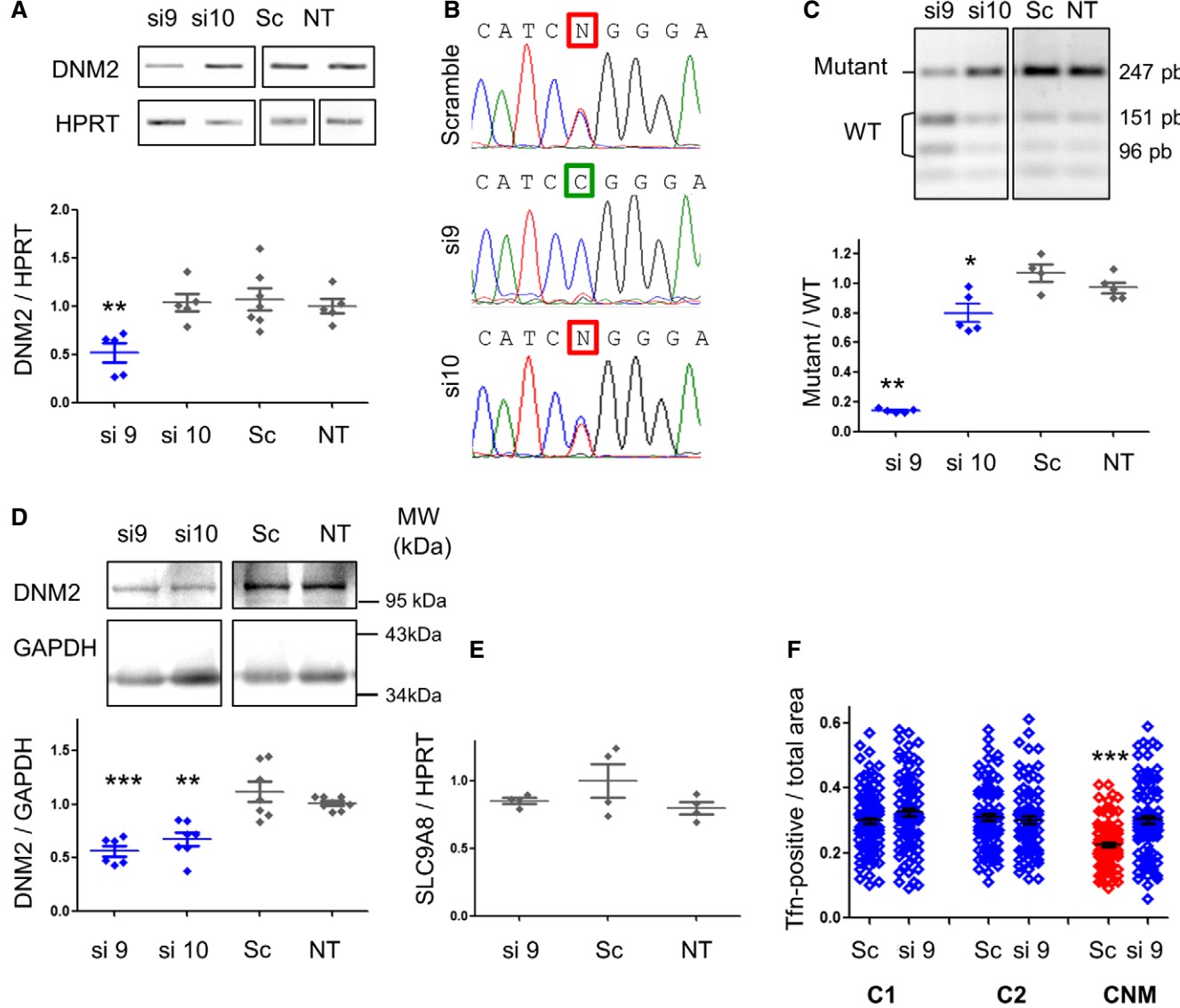

**Figure 7. Si9 exhibits allele-specific silencing and functional rescue in patient-derived fibroblasts.**

A   *DNM2* and *HPRT* RT–PCR products and quantification of *DNM2* expression normalized to *HPRT* ($n \geq 5$).
B   Sequence of *DNM2* amplicons from siRNA-transfected cells. Squares indicate the mutant nucleotide (N = T and C).
C   PfoI digestion profile of the *DNM2* amplicons and quantification of the mutant/WT ratio ($n = 5$).
D   DNM2 Western blot and quantification of signal by densitometry. GAPDH was used as loading control ($n = 7$).
E   Effect of si9 on SLC9A8 expression. Bars on scatter plot represent mean $\pm$ SEM of SLC9A8 expression normalized to HPRT. Statistical analysis was performed using a two-tailed Mann–Whitney *U*-test compared to scramble ($n = 4$).
F   15-min transferrin uptake in patient-derived fibroblasts ($n = 100$ cells for each cell line). Statistical analysis was performed using a Mann–Whitney *U*-test (***$P < 0.001$ versus the two control cell lines (C1 and C2) transfected with the same siRNA).

Data information: Scatter plot bars represent mean $\pm$ SEM. In (A, C and D), *$P < 0.05$, **$P < 0.01$, ***$P < 0.001$ using a one-tailed Mann–Whitney *U*-test compared to scramble. In (A–F), siRNAs were transfected at 50 nM for 48 h.

most frequent of them, that is the p.R465W mutation (30% of patients). We show here the proof of concept for AS-RNAi therapy for *DNM2*-related CNM in murine model and CNM patient-derived cells.

One remarkable facet of this approach, stemming from animal models, is the low therapeutic threshold which meant that total silencing of mutant mRNA is not absolutely required to reach therapeutic benefit *in vivo* (Xia *et al*, 2006; Loy *et al*, 2012; Jiang *et al*,

2013). In a knock-in mouse model of hypertrophic cardiomyopathy resulting from a dominant single nucleotide substitution in the *Myh6* gene, the disease was prevented by silencing only 30% of the mutant transcript (Jiang *et al*, 2013). Similarly, 3 months after the injection, sh9 normalizes muscle structure and function with a reduction of around 40–50% of the mutant *Dnm2* transcripts *in vivo*. Moreover, after 1 month, force improvement already takes place while the reduction in the *Dnm2* mutant is barely detectable.

Noteworthy, AAV1 exhibits a rapid onset but moderate level of transgene expression (Zincarelli *et al*, 2008). Thus, the incomplete restoration of muscle function observed after 1 month may result from submaximal level of transgene expression reached at this time or can reflect that the molecular and cellular processes necessary to achieve functional rescue require a longer time. However, these findings support a low therapeutic threshold and are of particular interest for future development of a similar strategy to target all CNM-associated *DNM2* mutations. Indeed, the siRNA-targeted sequences, fixed by the location of the mutation, may be located in regions not ideal for RNAi leading to difficulties to design optimal siRNA. A low therapeutic threshold will be a valuable asset in this context.

Previous *in vivo* studies in animal models also demonstrated essential features of AS-RNAi therapy including rapidity of therapeutic effects (Nobrega *et al*, 2013, 2014) and safety (Shukla *et al*, 2007; Rodriguez-Lebron *et al*, 2009; Jiang *et al*, 2013). Interestingly, this strategy is not only enabled to prevent the appearance and disease progression (reviewed in Trochet *et al*, 2015), but was also able to rescue a phenotype when applied to symptomatic animals (Nobrega *et al*, 2013). However, this last point may be a potential limitation for therapy since we and others (Jiang *et al*, 2013) have shown that allele-specific silencing was less efficient in reversing established disease. If the cause was not identified in the mouse model of cardiomyopathy (Jiang *et al*, 2013), our results point towards a decrease in AAV-mediated transduction capacity in heterozygous muscle when treated at 6 months of age which probably impedes therapeutic benefit. Given that there is no observed necrosis–regeneration cycle in the KI-*Dnm2* mice, the loss of viral genomes cannot be exclusively linked to muscle fibre loss as reported in dystrophic models or in wild-type mice after injury-induced regeneration (Le Hir *et al*, 2013). AAVs reach the nucleus by multiple intracellular trafficking events including binding to the cell's membrane, receptor-mediated endocytosis, trafficking through the endosomal system and endomembrane escape before nucleus entry (Nonnenmacher & Weber, 2012). Regarding known functions of DNM2, one can hypothesize that, when injected in a more severely affected muscle, defective endocytosis and/or trafficking over time leads to transduction inefficiency through decreased entry into the cell and/or abnormal targeting of the virion into degradative pathways. Improvement of AAV-mediated transduction, as already achieved in skeletal muscle by insulin treatment (Carrig *et al*, 2016) or removal of capsid phosphorylation sites (Qiao *et al*, 2010), may overcome this problem as well as an increase of the viral titre. Alternatively, despite several mechanisms shared with AAV for trafficking from the extracellular medium to the nucleus, non-viral delivery methods using cationic lipids (David *et al*, 2013) or nanoparticles (Davis *et al*, 2010) can be developed.

A suitable allele-specific siRNA for treatment of dominant inherited diseases, where WT and mutant alleles are similarly expressed, should reduce expression of target mRNA and protein around 50% resulting from silencing of the mutant allele without affecting the WT. The si9 identified in this study fulfils these criteria in patient-derived cells and in an animal model for CNM and represents a promising molecule for clinical perspectives. Behaviour of si10 is also of particular interest for a better understanding of AS-RNAi therapy. Compared to si9, si10 exhibits similar efficiency for allele-specific silencing *in vitro* in mouse cells but loses this property in

human cells and *in vivo* in mice. In addition, in human cells, si10 does not induce mRNA degradation but leads to protein content reduction in agreement with a possible shift towards a "microRNA effect" only affecting translation of the target as already shown for siRNA (Saxena *et al*, 2003). Overall, these features highlight absence of definitive rules for development of allele-specific molecules which remains largely dependent of empirical testing. In addition, our data argue for requirement of validation of allele-specific candidates in different models ideally including animal model of the disease and patient-derived cells.

Among other possible therapeutic strategies ongoing for CNM, a treatment with acetylcholinesterase inhibitor has been reported to lead to improvement of muscle strength in few DNM2-CNM patients (Gibbs *et al*, 2013). We have also previously succeeded to correct the *DNM2* mRNA by trans-splicing, while with a very low efficiency (Trochet *et al*, 2016). Targeting DNM2 expression is also of interest beyond the CNM due to *DNM2* mutations. Indeed, Tasfaout and collaborators succeeded to revert the phenotype of MTM1-myotubular myopathy mouse model by knock-down Dnm2 using anti-sense oligonucleotide (Tasfaout *et al*, 2017).

AS-RNAi is promising as a future therapy for AD-CNM patients, while identification and understanding of potentials off-target activities such as induction of inflammatory response and undesired gene silencing are an important consideration (Jackson & Linsley, 2010). In this study, we rule out interferon response in treated mice, undesired silencing on few off-target candidate genes exhibiting highest similarities with the targeted sequence in murine and human cells and toxicity in mouse muscle and human cells. However, extensive investigation of the potential off-targets of si9 in human and murine cells, especially by high-throughput screening, will be required in order to clarify the side-effects of this therapeutic strategy. The next steps of preclinical evaluation will also include assessment of long-term maintenance of therapeutic benefit over time in the KI-*Dnm2* mice, toxicity studies and impact of treatment in other tissues after systemic delivery. With these remarks, our pioneer proof of concept of allele-specific silencing for the most frequent *DNM2* mutation responsible for AD-CNM allows us to envision a treatment for this neuromuscular disorder and paves the way for the other CNM mutations as well as the other *DNM2*-linked inherited diseases.

# Materials and Methods

### Data analysis and statistics

Graphics and statistical analyses were performed with GraphPad Prism software version 5.00. Values were expressed as means ± SEM. The number of samples ($n$), representing the number of independent biological replicates, was indicated in the figure legends. For molecular analysis, the experiments were repeated at least three times per biological replicate and averaged. We used non-parametric statistical tests to analyse our data since the normality could not be assumed or tested ($n$ too small) or the variance was not equal between groups. Statistical comparisons between two groups were performed using unpaired one- or two-tailed Mann–Whitney *U*-test as specified. Statistical tests applied are indicated in the figure legends. $P < 0.05$ were considered as statistically significant. Individual *P*-values are available in Appendix Tables S3 and S4.

In most of the figures, the gels are cropped for more conciseness but samples presented were run on a same gel. Uncropped gels are shown in Appendix Figs S10 and S11.

## Cell cultures

Mouse embryonic fibroblasts (MEFs) were prepared from 13.5-day-old embryos. Cells were cultured at 37°C (5% $CO_2$) in Dulbecco's modified Eagle's medium (DMEM, Life Technologies, France) containing 10% foetal calf serum (FCS) supplemented with penicillin, streptomycin, L-glutamate and sodium pyruvate. Experiments were performed on MEFs in primary cultures, that is 2 or 3 passages after embryo dissection. Human skin fibroblasts from healthy control subjects (C1 and C2) and from one *DNM2*-linked CNM patient harbouring the p.R465W mutation were cultured using the same medium. CNM and control fibroblasts were immortalized using a lentiviral vector containing the sequence encoding the catalytic subunit of human telomerase (hTERT) as previously described (Aure *et al*, 2007). Mouse myoblast was immortalized using a lentiviral vector containing the sequence encoding CDK4. Mouse myoblast was plated on 1% matrigel-coated dishes and cultured in Dulbecco's modified Eagle's medium (DMEM, Life Technologies, France) containing 20% FCS supplemented with penicillin, streptomycin and 1% chicken embryo extracts.

For transfection, cells were grown to 70% confluency and transfected with siRNAs using JetPrime transfection reagent according to the manufacturer's protocol (Polyplus Transfection, France). Concentration of siRNAs for each experiment was indicated in corresponding figure legends. The mismatch position for the 12 siRNAs used in the *in vitro* screening was chosen based on previous studies (Trochet *et al*, 2015). Sequences of the siRNAs were indicated in Fig 1 and Appendix Table S1, and scramble siRNA (Eurogentec, Belgium) was used as control. Cells were harvested 48 h later for RNA and protein extraction or immunohistochemistry.

## Total RNA extraction and cDNA analysis

Total RNA was isolated from cells using RNA easy kit (Qiagen, France) according to the manufacturer's protocol. Cells were passed through a 22-G syringe several times for disruption in lysis buffer. Total RNA from muscle was isolated by TRIZOL reagent (Life Technologies, France) following standard protocol after disruption using Fastprep Lysing Matrix D and Fastprep apparatus (MP Biomedical, France). Total RNA (1 μg) was submitted to reverse transcription using the Superscript III reverse transcriptase kit (Life Technologies) and hexamer primers. The cDNAs were amplified by PCR under the following conditions: 96°C for 5 min, cycles of 30 s at 96°C, 30 s at the appropriate temperature (58–62°C), 30 s at 72°C and a final step of 7 min at 72°C. Semi-quantitative RT–PCR was used to determine the total *Dnm2* expression level, and the appropriate number of PCR cycles has been selected in order to have the amplification in the exponential range (i.e. 23 cycles to amplify Gapdh, 27 cycles for HPRT and 28 cycles for *Dnm2*). Sequences of the primers are indicated in Appendix Table S2. To quantify allele-specific silencing, an assay was developed using restriction enzymes allowing discrimination between wild-type and mutant cDNAs after digestion of the amplicons. The digestion of the human and murine *Dnm2* amplicons was performed after 32 cycles at the end of the exponential phase of amplification.

EcoNI was used in mouse (Appendix Fig S1) and PfoI (Appendix Fig S7) in human cells. For these assays, half of the PCR products was digested using 2 U of EcoNI (New England Biolabs, France) or PfoI (ThermoFisher Scientific, France) for 2 h at 37°C. Image acquisition of PCR products after agarose gel electrophoresis was performed using a Geni2 gel imaging system (Ozyme, France), and associated signal was quantified using ImageJ Software (NIH; http://rsbweb.nih.gov/ij). DNA sequencing was performed on 20 ng DNA/100 base pairs with 5 pmol of primers (Eurofins, France).

## Protein extraction and Western blot

Cell pellets and frozen TA muscles were homogenized in lysis buffer containing 50 mM of Tris–HCl pH 7.5, 150 mM NaCl, 1 mM EDTA, NP-40 1% supplemented with protease inhibitor cocktail 1% (Sigma-Aldrich, France). In addition, TA muscles were mechanically homogenized on ice in lysis buffer using a Potter-Elvehjem. After centrifugation (14,000 *g*, 4°C, 15 min), protein concentration in the supernatant was determined with the BCA Protein Assay Kit (Thermo Scientific Pierce, France). Twenty micrograms of proteins was mixed with loading buffer (50 mM Tris–HCl, SDS 2%, glycerol 10%, β-mercaptoethanol 1% and bromophenol blue) and denaturated at 90°C for 5 min. Protein samples were separated on SDS–PAGE 10% and transferred onto PVDF membranes (0.45 μm pore size, Life Technologies) overnight at 100 mA at 4°C. Page ruler prestained protein ladder (ThermoFisher Scientific, France) was used as molecular weight marker. Membranes were blocked for 1 h at room temperature in PBS containing non-fat dry milk 5% and Tween-20 0.1% and then exposed to rabbit polyclonal anti-Dynamin 2 antibody (1:400, Abcam ab3457, United Kingdom) or rabbit polyclonal anti-GAPDH antibody (1:2,000, Santa Cruz, France) in PBS–Tween-20 0.1%, milk 1% overnight at 4°C. Membranes were rinsed in PBS–Tween-20 0.1% and incubated 1 h with horseradish peroxidase-conjugated secondary antibody (1:20,000, anti-rabbit from Jackson ImmunoResearch, United Kingdom) in PBS–Tween-20 0.1%. Chemiluminescence was detected using ECL detection Kit (Merck-Millipore, Germany) in a G-Box imaging system (Ozyme, France), and signal quantification was performed using ImageJ software.

## AAV production

A cassette containing the small hairpin (sh) RNA under the control of H1 RNA polymerase III promoter or the muSEAP transgene under CMV promoter has been inserted in a pSMD2 expression plasmid. AAV vectors (serotype 1) were produced in HEK293 cells after transfection of the pSMD2-shRNA or -muSeap plasmids or empty pSMD2 plasmid, the pXX6 plasmid coding for viral helper genes essential for AAV production and the pRepCap plasmid (p0001) coding for AAV1 capsid as described previously (Riviere *et al*, 2006). Viral particles were purified on iodixanol gradients and concentrated on Amicon Ultra-15 100K columns (Merck-Millipore). The concentration of viral genomes (vg/ml) was determined by quantitative real-time PCR on a LightCycler480 (Roche diagnostic, France) by using TaqMan probe. A control pSMD2 plasmid was 10-fold serially diluted (from $10^7$ to $10^1$ copies) and used as a control to establish the standard curve for absolute quantification. Sequences of primers and probes are indicated in Appendix Table S2.

## Mice *and in vivo* transduction

The Dynamin 2 mutant mouse line was established on C57Bl/6 background at the Mouse Clinical Institute (MCI, Illkirch, France; http://www.ics-mci.fr/en/) (Durieux *et al*, 2010b). All mice used in this study were housed on a 12-h light/dark cycle and received standard diet and water *ad libitum* in the animal facility of the University Pierre et Marie Curie, Paris. Injections were performed under isoflurane anaesthesia. We chose male mice for the treatment since the characterization of the model has been done on male, the wild-type (wt) littermates were used as controls (Durieux *et al*, 2010b). Two intramuscular injections of 30 µl within 24-h interval were performed using 29-G needle in TA muscles. AAV-sh9, AAV-sh10 and AAV-control (empty AAV vector) were injected at $5 \times 10^{12}$ vg/kg at 1 month of age and $3.3 \times 10^{12}$ vg/kg at 6 months of age in order to achieve $10^{11}$ vg/muscle. For the 1M-1M series, eight heterozygous KI-*Dnm2*[R465W] males aged of 1 month were injected (5 TA per condition sh9, sh10 and control) and then analysed for the muscle force, histology and molecular biology. Five age-matched non-injected WT mice were used as control. For the 1M-3M series, 12 heterozygous KI-*Dnm2*[R465W] males aged of 1 month were injected (8 TA per condition sh9, sh10 and control) and analysed for the muscle force, but then, four were used for histology and four for molecular biology. Four age-matched non-injected WT mice were used as control. For the 6M-3M series, 12 heterozygous KI-*Dnm2*[R465W] males aged of 6 months were injected (12 TA per condition sh9 and control) and analysed for the muscle force, and then, six were used for histology and six for molecular biology. Three age-matched non-injected WT mice were used as control. For the AAV-muSEAP series, four heterozygous KI-*Dnm2*[R465W] (one male, three females) were injected at 1 month of age ($n = 8$ TA) in order to achieve $10^{11}$ vg/muscle and six heterozygous KI-*Dnm2*[R465W] (two males, four females) was injected at 6 months of age ($n = 12$ TA) in order to achieve either $10^{11}$ vg/muscle ($n = 6$) or $10^{12}$ vg/muscle ($n = 6$).

## Muscle contractile properties

The isometric contractile properties of TA muscles were studied *in situ* on mice anaesthetized with 60 mg/kg pentobarbital. The distal tendon of the TA muscle was attached to a lever arm of a servomotor system (305B Dual-Mode Lever, Aurora Scientific). The sciatic nerve was stimulated by a bipolar silver electrode using a supramaximal (10 V) square wave pulse of 0.1-ms duration. Absolute maximal isometric tetanic force was measured during isometric contractions in response to electrical stimulation (frequency of 25–150 Hz; train of stimulation of 500 ms). All isometric contraction measurements were made at optimal muscle length. Force is expressed in grams (1 g = 9.8 mNewton). Mice were sacrificed by cervical dislocation, and TA muscles were weighted. Specific maximal force was calculated by dividing absolute force by muscle weight.

## Histomorphological analyses

Tibialis anterior muscles were frozen in liquid nitrogen-cooled isopentane. Transverse sections of TA muscle (8 µm thick) were stained with haematoxylin and eosin (HE) and reduced diphosphopyridine nucleotide diaphorase staining (DPNH) by standard methods (Dubowitz & Sewry, 2007). The muSeAP detection on muscle

sections was performed on slices fixed with 0.5% glutaraldehyde. Slices were washed twice with PBS, and endogenous alkaline phosphatase was heat inactivated for 30 min at 65°C before a 5-h incubation at 37°C in 0.165 mg/ml 5-bromo-4-chloro-3-indolylphosphate and 0.33 mg/ml of nitroblue tetrazolium (NBT/BCIP, Promega) in 100 mM Tris–HCl, 100 mM NaCl and 50 mM MgCl$_2$. Sections were then counterstained with nuclear fast red, mounted and analysed by light microscopy. Light microscopy was performed using an upright microscope (DMR, Leica), and images were captured using a monochrome camera (DS-Ri1, Nikon) and NIS-Elements BR software (Nikon, France). Exposure settings were identical between compared samples and viewed at room temperature. For quantification of the frequency of fibre size, the maximum diameter was calculated for each fibre on TA muscle sections labelled with anti-laminin antibody (1:500, ab11575 Abcam, United Kingdom) using ImageJ software and the normal distribution of the values was plotted using Microsoft Excel software. Histomorphological abnormalities were counted on DPNH staining (two sections per animals) using ImageJ software.

## Immunocytochemistry

Cells and muscle tissue cryosections (8 µm thick) were fixed in paraformaldehyde 4% (15 min at room temperature). After washing in PBS, cells and cryosections were permeabilized in Triton X-100 0.5% in PBS for 10 min at room temperature and blocked in PBS–Triton X-100 0.1%, BSA 5% and donkey serum 5% for 30 min. Samples were incubated with rabbit anti-Dynamin 2 (1:400 (Abcam ab3457), rabbit anti-laminin (1:500, Abcam ab11575) or rabbit anti-Desmin (1:200, ab15200) overnight at 4°C, in PBS with Triton X-100 0.1% and BSA 1%. After PBS–Triton X-100 0.1% washes, samples were incubated with donkey anti-rabbit AlexaFluor488 secondary antibody (1:1,000, Life Technologies, France) for 60 min at room temperature. The slides were mounted with Vectashield mounting medium (Vector Laboratories, United Kingdom). Images were acquired using an axiophot microscope (Zeiss, France) or a macroscope (Nikon AZ100).

## Viral genome quantification

Transduction efficacy was evaluated through quantification of viral genomes in injected muscles. Genomic DNA was extracted from mouse muscle sections using DNA purification kit (Promega, France) according to the manufacturer's protocol. Copy numbers of AAV genomes were quantified on 100 ng of genomic DNA by quantitative real-time PCR on a LightCycler480 (Roche diagnostic, France) by using TaqMan probe. A control pSMD2 plasmid was 10-fold serially diluted (from $10^7$ to $10^1$ copies) and used as a control to establish the standard curve for absolute quantification. Sequence of primers also used for viral titration is indicated in Appendix Table S2.

## Transferrin uptake assay

Two healthy control cell lines (C1 and C2) and one patient-derived cell line expressing the p.R465W mutation were used 48 h after siRNA transfection. Cells were cultured in DMEM without FCS for 45 min at 37°C. AlexaFluor488-Transferrin (Life Technologies, France) was added at 40 µg/ml, and cells were incubated at 37°C for 15 min. Cells were washed three times with PBS and fixed in paraformaldehyde 4%

**The paper explained**

**Problem**

Autosomal-dominant centronuclear myopathy (AD-CNM) is a rare congenital myopathy exhibiting a wide clinical spectrum ranging from severe-neonatal to mild-adult forms. The disease is due to heterozygous mutations (mainly missense) in the *DNM2* gene encoding Dynamin 2, a large GTPase involved in endocytosis, intracellular trafficking and cytoskeleton network regulation. There is no curative treatment available for now. Therefore, we have evaluated the therapeutic potential of allele-specific silencing strategy to specifically knock down the mutant allele and alleviate the phenotype in a CNM knock-in mouse model and patient-derived fibroblasts both harbouring the most frequent mutation (p.R465W).

**Results**

We report the identification of an efficient siRNA able to specifically knock down the R465W *DNM2* mutant allele *in vitro* in mouse and human cells. This specific knockdown leads to functional restoration of the impaired endocytosis in mutant human fibroblasts. We assessed the efficacy of the treatment *in vivo* in the CNM mouse model by intramuscular administration of the corresponding AAV-small hairpin RNA. Early treatment in the disease's time-course restores muscle forces and mass with a moderate decrease of the mutant allele (~40%). More modest results were obtained with a late administration when the phenotype is more pronounced due to a weaker transduction capacity of the muscle.

**Impact**

Our study establishes the proof of concept of allele-specific silencing to correct the most frequent *DNM2* mutation responsible for AD-CNM. This is also the first evidence of functional rescue obtained both in CNM human cells and in mouse model. Consequently, allele-specific silencing is clearly identified as a promising therapeutic strategy for AD-CNM, allowing envisions of a treatment for this neuromuscular disorder and in extension to other *DNM2*-linked inherited diseases.

at room temperature for 15 min. Stacks of cell images (0.5 μm interval) were obtained using a Leica SP2 confocal microscope. Fluorescent-positive surface was quantified on stack projection using ImageJ software and normalized to the total cell surface.

### Study approval

Experiments on human cells were approved by the French Ministry of Higher Education and Scientific Research (approval no. AC-2013-1868) and conformed to the principles set out in the WMA Declaration of Helsinki and the Department of Health and Human Services Belmont Report. Written informed consent was obtained from all individuals and patients, prior to the study.

Animal studies conform to the French laws and regulations concerning the use of animals for research and were approved by an external Ethical committee (approval no. 00351.02 delivered by the French Ministry of Higher Education and Scientific Research).

**Expanded View** for this article is available online.

### Acknowledgments

We thank the Pitié-Salpêtrière Imaging Platform (PICPS) for light microscopy imaging facilities, the Penn Vector Core, Gene Therapy Program (University of Pennsylvania, Philadelphia, USA) for providing pAAV1 plasmid (p0001) and Stéphanie Buart for mouse embryonic fibroblast preparation. This work was supported by the Institut National de la Santé et de la Recherche Médicale (INSERM), the Association Institut de Myologie (AIM), the Université Pierre et Marie Curie (UPMC), the Centre National de la Recherche Scientifique (CNRS), and the Agence Nationale de la Recherche (grant ANR-14-CE12-0009 to M.B. and Young Researcher grant ANR-14-CE12-0001-01 to S.V.).

### Author contributions

DT and MBi conceived and designed the experiments. DT, BP, LJ, SBZ, MBe, CP, SL, KM, AF, SV and MBi performed the experiments. AR and JL generated immortalized myoblast. DT, BP, PG and MBi analysed the data. DT, SV and MBi wrote the paper. DT, BP, SB-Z, MBe, CP, SL, KM, AF, SV, PG and MBi read and approved the final version of the manuscript.

### Conflict of interest

The authors declare that they have no conflict of interest.

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
