## [Review Process File · EMBO Molecular Medicine]

Allele-specific silencing therapy for dynamin 2-related dominant centronuclear myopathy

Delphine Trochet, Bernard Prudhon, Maud Beuvin, Cécile Peccate, Stéphanie Lorain, Laura Julien, Sofia Benkhelifa-Ziyyat, Aymen Rabai, Kamel Mamchaoui, Arnaud Ferry, Jocelyn Laporte, Pascale Guicheney, Stéphane Vassilopoulos and Marc Bitoun

Review timeline:

Submission date:	05 May 2017
Editorial Decision:	22 June 2017
Revision received:	20 October 2017
Editorial Decision:	8 November 2017
Revision received:	14 November 2017
Accepted:	20 November 2017

Editor: Céline Carret

Transaction Report:

1st Editorial Decision

22 June 2017

Thank you for the submission of your manuscript to EMBO Molecular Medicine. We have now heard back from the three referees whom we asked to evaluate your manuscript.

You will see from the set of comments below that all three referees find the study to be of interest for EMBO Molecular Medicine. Nevertheless, they highlight a few issues that need to be addressed in the next final version of your article. Ref.1 for example, requests to extend the analysis to muscle cells. All three would like to see further explanations and an extended characterisation of the phenotypes post-treatment. More controls, clarifications and better data presentation are also itemised.

I look forward to receiving your revised manuscript.

***** Reviewer's comments *****

Referee #1 (Comments on Novelty/Model System):

What I think could be missing here is the use of muscle cells to test the allele-specific silencing. The authors used first mouse embryonic fibroblasts and later patient-derived fibroblasts to test/evaluate siRNA specific effect on mutated Dnm2. I suggest to include a part testing the system on human muscle cells, as they are the cells to be targeted in AD-CNM. I appreciate a possible difficulty in obtaining patient-derived muscle cells with this specific mutation, even though authors affirmed this is one of the most common mutation for AD-CNM. Anyway, the authors do have a murine model available and they could indeed at least isolate muscle cells from it and test the functionality of allele-specific silencing.

Referee #1 (Remarks):

Trochet et al. described an allele-specific silencing of Dnm2 mutation responsible for autosomal

dominant centronuclear myopathy (AD-CNM). siRNAs designed to specifically target the dynamin2 mutated allele were tested in vitro in MEFs and patient-derived fibroblasts and in vivo in a dynamin2-related CNM mouse model, suggesting a potential amelioration of the muscle phenotype. The work is well written, well described and of clear interest. Anyway, some points need to be addressed.

Major points:

1. What it is mainly missing here is the use of murine and human muscle cells to evaluate the effect of Dnm2 allele-specific silencing. The authors first used mouse embryonic fibroblasts and later patient-derived fibroblasts to evaluate siRNA specific effect on mutated Dnm2. These sets of data are surely valid but muscle cells are the relevant ones here. Authors do have a murine disease model available and they could indeed isolate murine muscle cells and test on them the functionality of allele-specific silencing. Lastly, they should test relevant siRNA on patient-derived human muscle cells.
2. Regarding the in vivo experiments, authors explained that the different response to AVV-si9 treatment seen between young (1M-3M) and old mice (6M-3M) -with old mice showing no effect- is a direct the result of poor AVV transduction in old mice compared to young ones. However, this does not explain why AVV-si9 was not efficient in reducing Dnm2 mutated even in young mice 1 month after treatment (1M-1M). Why reduction in Dnm2 mutated is seen only after three months (1M-3M) and not after one month (1M-1M)?
3. In the in vivo studies Trochet et al referred generically to "histological abnormalities" while they are only testing one histological abnormality, which is DPNH. Authors should remove the generic label "histological abnormalities" from the graph as not appropriate. They should also test some other typical histological abnormalities as nuclear centralization/nuclei mislocalization, predominance of type 1 fibres, calcium concentration and intracellular Dnm2 and dysferlin accumulation. Also, the pictures showed here do not convince me that KI-Dnm2R465W muscles are greatly affected; this may be due to quality of pictures as well.
4. Referring to Fig 2B, the authors stated that sequencing of si9- and si10-treated MEFs (Fig 2B) only detected WT Dnm2 allele when they actually showed a clear band for mutated Dnm2 in Fig 2C. Why did the sequencing not pick this up?

Minor points:

5. Authors should provide more background on how the mutations in Dnm2 cause the muscle phenotype and clarify the following point for a broader audience: do the mutations in Dnm2 lead to altered protein expression or to no protein expression at all? Is the mutated Dnm2 toxic?
6. Are the mice used in Fig 4 the same used for data in Fig 3? In case they are, were some animals excluded from analysis in Fig3?
7. Even though this paper does not affect the novelty and validity of the study presented by of Trochet et al, authors should mention and discuss a paper just recently published on Nature Communications "Antisense oligonucleotide-mediated Dnm2 knockdown prevents and reverts myotubular myopathy in mice " by Tasfaout et al. Also, the authors should mention possible other therapeutic strategies ongoing for CNM to give the reader a better context.

Referee #2 (Remarks):

In this manuscript, Trochet et al. described a novel gene therapy approach for the treatment of a DNM2 related autosomal dominant centronuclear myopathy by allele-specific silencing of mutant DNM2 gene. They found early treatment (1 month of age) of a knockin DNM2 (KI-Dnm2) disease mouse model rescued myopathy disease manifestation. However, late treatment (6 months of age) of KI-Dnm2 was less efficiency to restore the disease phenotype due to a weaker capability of virus transduction to older mice. A similar strategy was successfully applied to restore endocytosis functions in patient-derived fibroblast cells.

Experiments reported in this paper are generally well designed and properly controlled. They address an important question in the field, regarding whether targeting the root cause of autosomal dominant centronuclear myopathy could be a potential therapeutic approach. This study has the potential to be of interest for the large audience of myopathy research fields.

In my opinion, however, a few points would need to be addressed to fully support the authors' main conclusions.

Major points:

1. At 3 weeks of age, HTZ mice exhibit a robust decrease of contractile property without reduced muscle mass and disease pathology. Treatment time was selected at 1 month old HTZ mice. It would be interesting to test whether RNAi treatment is still working at month 2 or 3 when contractile property, muscle mass and pathology are all affected.
2. How long will RNAi rescue effect last? The author should provide the follow-up functional study (at least additional 3 months) for 1 month treated mice.

Minor points:

1. The authors should show the efficiency of AAV1 (AAV1-EGFP) for TA muscles at 10^{11} vg/TA muscles when injected at 1 month. How is EGFP distribution in muscle tissue sections?
2. The total Dnm2 protein was drastically reduced by si9 and si10 in Figure 2D. Will wildtype Dnm2 be affected by si9 and si10 through miRNA mechanisms? If so, reduction of wild type Dnms influenced the phenotype?
3. The alleles can be distinguished by SNPs. It would be interesting to have a rescue experiment for patient-derived cells through allele-specific RNAi targeting DNM2 linked SNPs.
4. The authors should provide a possible solution for low AAV1 transduction for 6 month old mice. How about increase the virus titer? How about other AAV serotypes, like AAV6, 8, 9? How is EGFP distribution in muscle tissue sections?
5. The authors should explain whether muscle stem or progenitor cells could be involved in animal experiments.

Referee #3 (Remarks):

In this study, the authors show proof-of-principle of allele-specific RNA interference for autosomal dominant centronuclear myopathy due to a mutation in dynamin2 gene, both in a mouse model and in patient-derived fibroblasts. Their results suggest that even if the silencing of the mutant allele is not complete, there is enough functional effect to restore muscle structure and function in the mouse and human cells.

Major comments:

- 1) The number of mice evaluated in Figure 2, Figure 3 and Figure 6 is not the same. Molecular analysis of all animals used in the functional-structural evaluation would strengthen the results obtained concerning the allele specific silencing as well as the assessment of vg/nucleus in Figure 6.
- 2) The choice of the control AAV without shRNA sequence is questionable. This is a control for effect of transduction; however, a control for the silencing effect with a scramble AAV, as used in MEFs, is missing in vivo.
- 3) The effect of 3-months-treatment in older Dnm2-KI mice was only very mild. The phenotype of the mice at this age is more pronounced and could affect processes as viral endocytosis, which could lead then to reduced transduction efficiency. However no experiments with higher dose of virus have been performed.

Minor comments:

- 1) Authors should harmonize the figures for font and style used, verify that there is a space between number and its units, consistently use either "mutant" or "mutated". In addition, according to international rules, name of a gene is always written in italic (Dnm2 for mouse and DNM2 for human). The same is true for mRNA. Authors should harmonize it in the text.
- 2) Statistical analysis (when the same test is used) should be written only once per figure by the end of the legend. Authors should check for number of stars and their meaning (* P <0.05; ** P <0.01 and *** P <0.001).
- 3) The precise position of the murine mutation in the cDNA should be given at the beginning of Results (as it is done later for the human sequence). It is interesting that the nucleotide changed by the mutation is not the same in the mouse and human sequence even though is the first nucleotide of the codon 465. Authors might comment on that.
- 4) Serotype used is not specified in the Results as well as the kind of injection used in the mice. Why did the authors inject two times in 24 hours? Was two-time the same dose? Did the author check whether neutralizing antibodies were produced?

- 5) Figure 2: In panel D the size of the proteins could be given. In the legend "Mut:mutant" is specified but the abbreviation is not used in the figure.
- 6) Figure 3: WT muscles should be added in panel B to see the wild-type nucleotide and in panel C as control of the EcoNI digestion profile. Why after PCR and EcoNI digestion there is a stronger upper band for the WT amplicon?
- 7) Figure 4: It is more logical to show first the histological staining of the muscles and then the quantification. What is the unit of absolute Force? g for grams? Is that more common than (m)Newton unit for Force?
- 8) Figure 5: Quantification of histological abnormalities are missing.
- 9) Figure 6: The units of the y-axis in panel A are missing. Are the expression data related to control or WT? The Vg/nucleus is highly variable in the different mice (n=4). Are these the same mice as shown in figure 3C? What is the interpretation of the author for the same effect in silencing but very different number of vg/nucleus?
- 10) Potential silencing off-targets evaluation in MEFs or in vivo could be shown.
- 11) Supplemental Figure 4: Why is the percent of histological abnormalities in HTZ control mice in the group 1M+1M higher than the HTZ control mice in the group 1M+3M? Isn't the accumulation of DPNH a marker for progress of the disease?
- 12) Supplemental Figure 7: It is not clear why the mutant/WT ratio have been calculated using BglI digestion profile (and not PfoI as shown before). The size of the digested products should also be given. In panel C the density of non-transfected fibroblasts is higher than after transduction with si9 and scramble. Were the NT cells subjected to same procedure (transfectant w/o si9 or scramble)? Was the cell density after transduction with lower concentration of si9/scramble higher?

1st Revision - authors' response

20 October 2017

Referee #1:

Trochet et al. described an allele-specific silencing of Dnm2 mutation responsible for autosomal dominant centronuclear myopathy (AD-CNM). siRNAs designed to specifically target the dynamin2 mutated allele were tested in vitro in MEFs and patient-derived fibroblasts and in vivo in a dynamin2-related CNM mouse model, suggesting a potential amelioration of the muscle phenotype. The work is well written, well described and of clear interest. Anyway, some points need to be addressed.

Major points:

1. What it is mainly missing here is the use of murine and human muscle cells to evaluate the effect of Dnm2 allele-specific silencing. The authors first used mouse embryonic fibroblasts and later patient-derived fibroblasts to evaluate siRNA specific effect on mutated Dnm2. These sets of data are surely valid but muscle cells are the relevant ones here. Authors do have a murine disease model available and they could indeed isolate murine muscle cells and test on them the functionality of allele-specific silencing. Lastly, they should test relevant siRNA on patient-derived human muscle cells.

Answer: We do not have human muscle cells harboring the R465W mutation therefore, in agreement with reviewer's recommendation, we chose to test the functionality of the strategy on immortalized mouse myoblasts derived from the Knock-in model. The results confirmed the specificity of the silencing on the mutant allele in myoblasts cells.

Changes made in the manuscript:

We added a new supplemental figure 3 (see below) showing these results and mention them at the ends of the first result section. We described the protocol in the material and methods section.

Page 7; lane 124:

...and protein levels by half in HTZ MEF. The ability of 100nM of si9 and si10 to specifically silence the mutant allele was confirmed by RT-PCR in immortalized mouse myoblasts derived from heterozygous (HTZ) KI-Dnm2 (Appendix Figure S3).

2. Regarding the in vivo experiments, authors explained that the different response to AVV-si9 treatment seen between young (1M-3M) and old mice (6M-3M) -with old mice showing no effect- is a direct the result of poor AVV transduction in old mice compared to young ones. However, this does not explain why AVV-si9 was not efficient in reducing Dnm2 mutated even in young mice 1 month after treatment (1M-1M). Why reduction in Dnm2 mutated is seen only after three months (1M-3M) and not after one month (1M-1M)?

Answer: The reduction of Dnm2 mutant allele is actually already seen after 1 month treatment while indeed more moderate than after 3 months (~25% decrease versus 50%) in accordance with the consequences on muscle phenotype. We do not have definitive answer to explain that difference; however latency for all the molecular and cellular processes to take place may be a hypothesis.

We discuss this point page 12, lane 259: "...after 1 month force improvement already takes place while the reduction of the Dnm2 mutant is barely detectable. Noteworthy, AAV1 exhibits a rapid onset but moderate level of transgene expression (Zincarelli et al. 2008). Thus, the incomplete restoration of muscle function observed after 1 month may result from submaximal level of transgene expression reached at this time or can reflect that the molecular and cellular processes necessary to achieve functional rescue require a longer time. "

3. In the in vivo studies Trochet et al referred generically to "histological abnormalities" while they are only testing one histological abnormality, which is DPNH. Authors should remove the generic label "histological abnormalities" from the graph as not appropriate. They should also test some other typical histological abnormalities as nuclear centralization/nuclei mislocalization, predominance of type 1 fibres, calcium concentration and intracellular Dnm2 and dysferlin accumulation.

Answer: As requested we replaced histological abnormalities by "histological abnormality" in the graphs, legend and text. We understand the reviewer point, unfortunately nuclear centralization/nuclei mislocalization and predominance and atrophy of type 1 fibres are morphological features of DNEM2-related CNM in human, they are not observed in the mouse model.

Finally, the elevated cytosolic Ca^{2+} concentration and the cytoplasmic retention of DNEM2 and dysferlin were evidenced after several days in culture of HTZ isolated muscle fibers derived from

the flexor digitorum brevis muscle. We don't know if comparable defaults would be present in the tibialis anterior muscle, furthermore the culture of isolated fibres derived from tibialis anterior muscle is somewhat arduous experiment.

4. Also, the pictures showed here do not convince me that KI-Dnm2R465W muscles are greatly affected; this may be due to quality of pictures as well.

Answer: Our aim was to present pictures representative of the counting. Indeed, in Figure 4D DPNH picture presented for the heterozygous control contains 19/68 fibres with a central anomaly (i.e. 30%) the heterozygous treated with sh10 7/61 (i.e. 11%).

In Figure 5C, picture presented for 6 months heterozygous control contains 11/44 fibres with a central anomaly (i.e. 25%) the heterozygous treated with sh9 4/39 (i.e.10%). We hope that the higher resolution provided for the revised version will solve this issue.

5. Referring to Fig 2B, the authors stated that sequencing of si9- and si10-treated MEFs (Fig 2B) only detected WT Dnm2 allele when they actually showed a clear band for mutated Dnm2 in Fig 2C. Why did the sequencing not pick this up?

Answer: The sequencing was performed on amplicons obtained after semi-quantitative PCR (in the exponential phase of the amplification) (i.e 28 cycles for Dnm2). Conversely, the EcoNI digestion was performed on amplicon from PCR at the end of the exponential phase of amplification (i.e 32 cycles) to better detect all digested products. We agree that this point need clarification and we added more explanation in the text and material and methods sections.

Changes made in the manuscript:

- In the Figure 2 legend: Figure 2. (A) Semi-quantitative Dnm2 and Gapdh RT-PCR products and quantification of Dnm2 expression normalized to Gapdh.
- In material and method, (page 18, line 374) "Semi-quantitative RT-PCR was used to determine the total dnm2 expression level, the appropriate number of PCR cycles has been selected in order to have the amplification in the exponential range (i.e. 23 cycles to amplify Gapdh, 27 cycles for HPRT and 28 cycles for Dnm2). Sequences of the primers were indicated in Appendix Table S2. To quantify allele-specific silencing an assay was developed using restriction enzymes allowing discrimination between wild-type and mutated cDNAs after digestion of the amplicons. The digestion of the human and murine Dnm2 amplicons was performed at the end of the exponential phase of amplification (i.e 32 cycles) to better detect all digested products

Minor points:

6. Authors should provide more background on how the mutations in Dnm2 cause the muscle phenotype and clarify the following point for a broader audience: do the mutations in Dnm2 lead to altered protein expression or to no protein expression at all? Is the mutated Dnm2 toxic?

Answer: We have added a paragraph at the end of the introduction section giving more information about consequences of *DNM2* mutations.

Changes made in the manuscript:

- page 4, line 65:
- "The DNEM2 protein is ubiquitously expressed and to date, there is no explanation for the tissue-specific impact of the *DNM2*-mutations. Several Dnm2 dependent processes have been shown to be impaired by CNM mutations and supposed to contribute to muscle pathophysiological mechanisms (i.e. endocytosis, microtubules network, T-tubules organization and recently actin-mediated trafficking) (A.-C. Durieux et al. 2010; González-jamett et al. 2017). In mouse muscle fiber, Dnm2 presents a striated transversal staining pattern on the I-band of the sarcomere centered on the Z-line. Dnm2 localized to the perinuclear MTOC, Golgi apparatus, microtubules, sarcoplasmic reticulum, is enriched at the neuromuscular junction and colocalize with clathrin heavy chain (CHC) (A. C. Durieux et al. 2010). *DNM2* mutations in AD-CNM patients are mostly missense (Bohm hum mut 2012) and when tested the mutant protein is expressed normally (Bitoun et al. 2005; Bitoun et al. 2009). Mutations are thought to be responsible for a gain of function and/or a dominant negative effect through an increased GTPase activity and formation of abnormal stable Dnm2 oligomers (Wang et al. 2010; Kenniston and Lemmon 2010).. "

7. Are the mice used in Fig 4 the same used for data in Fig 3? In case they are, were some animals excluded from analysis in Fig3?

Answer: Figure 4 represents forces and histological measurements in tibialis anterior 3 months after the injection of 1 month old mice. For these experiments 8 tibialis anterior have been injected for each condition (*i.e.* sh9, sh10 and control). At the time of sacrifice, mass and force measurements have been performed on all injected tibialis anterior. Then, 4 TA were sliced for histological analysis and 4 were used for molecular analysis, which explains the discrepancies in the number of animals. Similar protocol was followed for the 6-3M series. In order to clarify this point we added a paragraph in the material and method section.

Changes made in the manuscript:

In the material and method section, Page 21, lanes 426-431:

“For the 1M-1M series, 5 TA were injected for each condition (*i.e.* sh9, sh10 and control) and then analyzed for the muscle force, histology and molecular biology. For the 1M-3M series, 8 TA were injected for each condition (*i.e.* sh9, sh10 and control) and analyzed for the muscle force, and then 4 were used for histology and 4 for molecular biology. For the 6M-3M series 12 TA were injected for each condition (*i.e.* sh9 and control) and analyzed for the muscle force, and then 6 were sliced for histology and 6 were used for molecular biology.”

8. Even though this paper does not affect the novelty and validity of the study presented by of Trochet et al, authors should mention and discuss a paper just recently published on Nature Communications "Antisense oligonucleotide-mediated Dnm2 knockdown prevents and reverts myotubular myopathy in mice " by Tasfaout et al. Also, the authors should mention possible other therapeutic strategies ongoing for CNM to give the reader a better context.

Answer: We have added the following section in the discussion page 15:

“Among other possible therapeutic strategies ongoing for CNM a treatment with acetylcholinesterase inhibitor has been reported to lead to improvement of muscle strength in few DNM2-CNM patients (Gibbs et al. 2013). We have also previously succeeded to correct the DNM2 mRNA by trans-splicing, while with a very low efficiency (Trochet et al. 2016). Targeting DNM2 expression is also of interest beyond the CNM due to *DNM2* mutations. Indeed, Tasfaout and collaborators succeeded to revert the phenotype of MTM1-myotubular myopathy mouse model by knockdown Dnm2 using antisense oligonucleotide (Tasfaout et al. 2017).

Referee #2 (Remarks):

In this manuscript, Trochet et al. described a novel gene therapy approach for the treatment of a DNM2 related autosomal dominant centronuclear myopathy by allele-specific silencing of mutant DNM2 gene. They found early treatment (1 month of age) of a knockin DNM2 (KI-Dnm2) disease mouse model rescued myopathy disease manifestation. However, late treatment (6 months of age) of KI-Dnm2 was less efficiency to restore the disease phenotype due to a weaker capability of virus transduction to older mice. A similar strategy was successfully applied to restore endocytosis functions in patient-derived fibroblast cells.

Experiments reported in this paper are generally well designed and properly controlled. They address an important question in the field, regarding whether targeting the root cause of autosomal dominant centronuclear myopathy could be a potential therapeutic approach. This study has the potential to be of interest for the large audience of myopathy research fields.

In my opinion, however, a few points would need to be addressed to fully support the authors' main conclusions.

Major points:

1. At 3 weeks of age, HTZ mice exhibit a robust decrease of contractile property without reduced muscle mass and disease pathology. Treatment time was selected at 1 month old HTZ mice. It would be interesting to test whether RNAi treatment is still working at month 2 or 3 when contractile property, muscle mass and pathology are all affected.

Answer: The aim of the late treatment was precisely to check whether the treatment is working on a pronounced phenotype. Our results showed a lower efficiency of the treatment in older mice and also point on transduction deficiency. Therefore, we indeed don't know whether the treatment would work on context of efficient transduction.

We agree with the reviewer that it would be important to test the treatment at intermediate stages of the disease progression. However, we think that it now appears important to first characterize and evaluate the progression of the transduction defaults in these mice before testing the treatment. This will allow a better interpretation of the results and an adaptation of the therapeutic strategy.

To give first answer to that question we looked at transduction efficiency in a group of 2 month old mice compared with groups of 1 and 6 months old mice. Mice were injected for one month with AAV serotype 1, encoding the *muSEAP* reporter gene (dose 10^{11} vg/TA similar to the one used previously). We then evaluated and compared transduction efficiency through quantification of the amount of viral genomes per ng of DNA.

As showed in the figure below the transduction default is already present at 2 months age. Therefore we can hypothesize that a similar treatment to the one performed in our study at one month old would be less efficient in 2 months old mice.

Please note that we did not include the results at 2 months old in the revised version of the paper for more conciseness.

2. How long will RNAi rescue effect last? The author should provide the follow-up functional study (at least additional 3 months) for 1 month treated mice.

Answer: We agree that the issue of the duration of effectiveness of shRNA therapy is of primary importance and we included it as next step of the preclinical evaluation (see discussion page 16, lane 3. Regarding others studies, it has been shown that the benefits of allele specific silencing of the Myh6 mutant in the mouse model of hypertrophic Cardiomyopathy dissipated over time (from 11 months) (J. Jiang et al. 2013) while a long-term therapeutic effect of shRNA was observed in cone-rod dystrophy mouse models (up to 12 months) (L. Jiang et al. 2011). To date no information from a similar treatment on skeletal muscle is available. However, as we think it is important to have at least 12 months of follow-up, the results will not be available in reasonable time for the paper revision and we apologize for that.

Minor points:

3. The authors should show the efficiency of AAV1 (AAV1-EGFP) for TA muscles at 10^{11} vg/TA muscles when injected at 1 month. How is EGFP distribution in muscle tissue sections?

Answer: To answer this question we have performed a new series of experiments including the injection of heterozygous 1 month old mice with AAV serotype 1, encoding the *muSEAP* reporter gene at the same dosage than the one used for the treatment (i.e. 10^{11} vg/TA).

As illustrated below, one month after the injection histochemical detection of muSEAP shows an expression in all muscle fibre. This new result has been included in the result section page 10, lane

201 «Consistently, the phosphatase activity is detected in all muscle fibres at one month…» and as a Figure 6D.

4. The total Dnm2 protein was drastically reduced by si9 and si10 in Figure 2D. Will wildtype Dnm2 be affected by si9 and si10 through miRNA mechanisms? If so, reduction of wild type Dnms influenced the phenotype?

Answer: As discussed page 15 lane 302, we considered as a possibility a shift toward a “microRNA effect” affecting only translation of the target for the si10 in human cells. Indeed, we observed protein content reduction with no mRNA reduction (see Figure 7D). However, we do not know whether this microRNA effect affects specifically one allele since the protein content stays above 50%.

As noticed by the reviewers in MEF cells Figure 2 D, the protein content may be indeed reduced “slightly” below 50% while only the mutant allele is impacted at mRNA level (figure 2C and appendix 2A). Therefore, the wild type Dnm2 allele might be possibly affected by si9 or si10 RNA through miRNA mechanisms. Supporting this idea, it is known that partial complementarity of a siRNA with the targeted mRNA leads to inhibition of translation rather than RNA cleavage (Hu, Liu, and Corey 2010; Carthew and Sontheimer 2009). However, moderate diminution of the wild type allele would be well tolerated as there is no evidence of Dnm2 haploinsufficiency in human and the hemizygous Dnm2 mice are healthy. In addition, our *in vivo* results corroborate this presumption since the phenotype is improved with the corresponding sh9 and sh10 RNA.

5. The alleles can be distinguished by SNPs. It would be interesting to have a rescue experiment for patient-derived cells through allele-specific RNAi targeting DN2 linked SNPs.

Answer: We agree with the reviewer that it would be interesting to have a rescue experiment through allele-specific RNAi targeting a mutant associated SNP. Indeed, this strategy has been already nicely used in others disease in particular in the case of triplet expansion disease and would be especially appropriate considering the large DN2 mutational spectrum associated with CNM. However, we aimed at establishing here the proof of concept for the present approach on the most frequent R465W mutation, since validation was feasible on the mouse model as well as available patient-derived cells. Development of similar approach targeting SNP, which will be certainly include as a part of our future developments for this therapeutic strategy, is largely out of the scope of the current study and will require specific technological developments.

6. The authors should provide a possible solution for low AAV1 transduction for 6 month old mice. How about increase the virus titer? How is EGFP distribution in muscle tissue sections?

Answer: To answer these questions we have injected 6 months old heterozygous mice with AAV serotype 1, encoding the *muSEAP* reporter gene at the dose used for the sh-RNA treatment (10^{11} vg/TA) and with a tenfold higher dose (10^{12} vg/TA). We have compared transduction efficiency through quantification of the amount of viral genomes and muSEAP distribution in muscle tissue section one month later.

The transduction efficiency is greatly improved in 6 months old mice by increasing the viral dose as shown in the Figure below. Indeed, the number of viral genome per ng after one month is significantly higher in mice injected with the high titer (A, $p < 0.05$, Two-sided Mann Whitney). Moreover, while muSEAP coloration is not detected in large area of muscle section in tibialis injected at the lower dose, the large majority of fibres expressed the phosphatase at the high dosage

(see B). Therefore increase the viral titer might be a solution to improve treatment efficiency in old mice.

This new result has been included in the result section page 10, lane 206 « .. In the same time, the effect of a ten-fold increase in the virus titer on the transduction in old mice was evaluated. The transduction efficiency was greatly improved in 6 months old mice by increasing the viral dose, since the number of vg/ng (8300 vg/ng) was comparable to the young mice and the large majority of fibres expressed the phosphatase (Figure 6 C and D)....” and as a Figure 6C.

7. How about other AAV serotypes, like AAV6, 8, 9?

Answer: To our opinion, taking into account the DNMT2 function, we believe that the origin of the transduction defect probably involves steps of the general process of AAV trafficking, regardless of the serotype. Consequently, we are not convinced that the use of another serotype would help.

8. The authors should explain whether muscle stem or progenitor cells could be involved in animal experiments.

Answer: To our knowledge, AAV do not efficiently transduce muscle satellite cells *in vivo*. For example, it has been shown that AAV1 genome is lost from Dystrophic muscle after regeneration (Le Hir et al. 2013). However, this would have no impact for the current treatment since there is no necrosis-regeneration cycle in human or mouse CNM muscle.

Referee #3 (Remarks):

In this study, the authors show proof-of-principle of allele-specific RNA interference for autosomal dominant centronuclear myopathy due to a mutation in dynamin2 gene, both in a mouse model and in patient-derived fibroblasts. Their results suggest that even if the silencing of the mutant allele is not complete, there is enough functional effect to restore muscle structure and function in the mouse and human cells.

Major comments:

1. The number of mice evaluated in Figure 2, Figure 3 and Figure 6 is not the same.

Answer: The number of mice evaluated in Figure 4 (not 2), Figure 3 and Figure 6 is indeed not the same. Figure 4 represents force and histological measurements in tibialis anterior 3 months after the injection of 1 month old mice. For these experiments 8 tibialis anterior have been injected for each condition (*i.e.* sh9, sh10 and control). At the time of sacrifice, mass and force measurements have been performed on all injected tibialis anterior. Then, 4 TA were sliced for histological analysis and 4 were used for molecular analysis, which explains the discrepancies in the number of animals. Therefore, mice presented in Figure 4A, E and F (Forces and mass) include the mice presented in Figure 3 (molecular) and in figure 4C and D (histology). Similar protocol was followed for the 6-3M series. In order to clarify this point we have added a paragraph in the material and method section.

Changes made in the manuscript:

In the material and method section, Page 21:

“For the 1M-1M series, 5 TA were injected for each condition (*i.e.* sh9, sh10 and control) and then analyzed for the muscle force, histology and molecular biology. For the 1M-3M series, 8 TA were injected for each condition (*i.e.* sh9, sh10 and control) and analyzed for the muscle force, and then 4 were used for histology and 4 for molecular biology. For the 6M-3M series 12 TA were injected for each condition (*i.e.* sh9 and control) and analyzed for the muscle force, and then 6 were sliced for histology and 6 were used for molecular biology.”

2. Molecular analysis of all animals used in the functional-structural evaluation would strengthen the results obtained concerning the **allele specific silencing** as well as the assessment of vg/nucleus in Figure 6.

Answer: Concerning the allele specific silencing we already get significant results in most cases despite a small n and the use of robust statistical test (*i.e.* non-parametric). Therefore increase the “n” will not change the conclusions. For the assessment of viral particles, we have performed a series of new experiments with the intramuscular injection of AAV1-muSEAP, and the “n” was increased (up to n = 6 or 8).

3. The choice of the control AAV without shRNA sequence is questionable. This is a control for effect of transduction; however, a control for the silencing effect with a scramble AAV, as used in MEFs, is missing *in vivo*.

Answer: We fully understand the reviewer’s concern, however the use of AAV-scramble as control is especially important to perform extensive study such as off-target effects. We considered the use of transduction control alone acceptable in the context of the results presented in the paper (*i.e.* no large expression data, which can indeed be influenced by activation of the RNAi machinery).

4. The effect of 3-months-treatment in older Dnm2-KI mice was only very mild. The phenotype of the mice at this age is more pronounced and could affect processes as viral endocytosis, which could lead then to reduce transduction efficiency. However no experiments with higher dose of virus have been performed.

Answer: To answer that question also raised by reviewer 2, we have performed a new series of experiments. We have injected the tibialis anterior of 6 months old heterozygous mice with AAV serotype 1, encoding the *muSEAP* reporter gene at the dose used for the treatment (10^{11} vg/TA) and with a tenfold higher dose (10^{12} vg/TA). We then compared transduction efficiency through quantification of the amount of viral genomes and muSEAP distribution in muscle tissue section one month later.

The transduction efficiency is greatly improved in 6 months old mice by increasing the viral dose as shown in the Figure below. Indeed, the number of viral genome per ng after one month is significantly higher in mice injected with the high dose. Moreover, while muSEAP coloration is not detected in large area of muscle section in tibialis injected at the lower dose, the large majority of fibres expressed the phosphatase at the high dosage (see B). Therefore increase the viral titer might be a solution to improve treatment efficiency in old mice.

This new result has been included in the result section page 10, lane 206 « .. In the same time, the effect of a ten-fold increase in the virus titer on the transduction in old mice was evaluated. The transduction efficiency was greatly improved in 6 months old mice by increasing the viral dose, since the number of vg/ng (8300 vg/ng) was comparable to the young mice and the large majority of fibres expressed the phosphatase (Figure 6 C and D)....” and as a Figure 6C.

**Minor comments:**

5. Authors should harmonize the figures for font and style used, verify that there is a space between number and its units, consistently use either "mutant" or "mutated". In addition, according to international rules, name of a gene is always written in italic (*Dnm2* for mouse and *DNM2* for human). The same is true for mRNA. Authors should harmonize it in the text.

Answer: It has been checked and modified.

6. Statistical analysis (when the same test is used) should be written only once per figure by the end of the legend. Authors should check for number of stars and their meaning (* $P < 0.05$; ** $P < 0.01$ and *** $P < 0.001$).

Answer: It has been done.

7. The precise position of the murine mutation in the cDNA should be given at the beginning of Results (as it is done later for the human sequence). It is interesting that the nucleotide changed by the mutation is not the same in the mouse and human sequence even though is the first nucleotide of the codon 465. Authors might comment on that.

Answer: It has been done, page 6 line 118 :”In the KI-*Dnm2*R465W mouse model, the missense mutation corresponds to a single point mutation in exon 11 (c.1393 A>T, p.R465W)”.

Codons CGG and AGG both encode Arginine, sometimes the genetic code redundancy occurs on the first nucleotide of the codon (it is also the case for Leucine and Serine)

In human: c.1393 C>T, pR465W codon CGG (Arg)>TGG (Tryp)

In Mouse: c.1393 A>T, pR465W codon AGG (Arg)>TGG (Tryp)

8. Serotype used is not specified in the Results as well as the kind of injection used in the mice.

Answer: Serotype and kind of injection have been added in the result section.

Page 7 line 151: “Adeno-associated virus (AAV) **serotype 1** vectors containing small hairpin (sh) RNA corresponding to si9 and si10..”

And line 155: “AAVs were injected *intramuscularly* at 10¹¹ viral genomes (vg)/TA muscle of HTZ KI-*Dnm2* mice at 1 month of age”

9. Why did the authors inject two times in 24 hours? Was two-time the same dose? Did the author check whether neutralizing antibodies were produced?

Answer: We performed the injection in two times (with the same dose) to allow the injection of larger amount of AAV (useful when the virus titer is low) and to minimize the risk of mis-injection on the side to the muscle.

We did not check the presence of neutralizing antibodies, since the second injection was performed before the 24th hours after the first injection meaning in a time window where the immune response is not stabilized (see Lorain et al. 2008).

10. Figure2: In panel D the size of the proteins could be given. In the legend "Mut:mutant" is specified but the abbreviation is not used in the figure.

Answer: Molecular weight of Dynamin 2(98kDa) and Gapdh (37kDa) has been added on figure 2 and 7. Mut: mutant has been removed from the legend

11. Figure3: WT muscles should be added in panel B to see the wild-type nucleotide and in panel C as control of the EcoNI digestion profile.

Answer: As requested we have added the wild-type EcoNI digestion profile as control in panel C. We are sorry we do not have the electropherogram corresponding to the wild-type semiQ RT-PCR presented in A.

12. Why after PCR and EcoNI digestion there is a stronger upper band for the WT amplicon?

Answer: The upper band of the WT amplicon digestion is more intense because it is larger (247 pb vs 146) and therefore bind more Ethidium Bromide.

13. Figure 4: It is more logical to show first the histological staining of the muscles and then the quantification.

Answer: It has been modified in Figure 4 and Appendix Figure S5.

14. What is the unit of absolute Force? g for grams? Is that more common than (m)Newton unit for Force?

Answer: The unit we used for absolute force is indeed g for grams (full name: gram force) it has been specify in the material and methods section page 20, lane 425. Newton is the SI unit and we specify the conversion in material and methods.

The conversion is as follows: $1 \text{ newton [N]} = 101.971621297793 \text{ gram [g]}$
 $1 \text{ gram [g]} = 9.8 \text{ m[N]}$

15. Figure 5: Quantification of histological abnormalities are missing.

Answer: Quantification of histological abnormalities has been added to Figure 5.

16. Figure 6: The units of the y-axis in panel A are missing. Are the expression data related to control or WT?

Answer: The missing unit was added to the graph.
 The data are expressed relative to the control (which is not a scramble)

17. The Vg/nucleus is highly variable in the different mice (n=4).

Answer: Quantification of the number of vg after AAV transduction is often quite variable. We haven't definitive explanation for this observation. This might be attributed to a non-homogeneous spreading of the AAV along muscle associated with DNA extraction from a little fraction of the muscle.

18. Are these the same mice as shown in figure 3C? What is the interpretation of the author for the same effect in silencing but very different number of vg/nucleus?

Answer: The mice used for viral quantifications are the ones used for histology and anomaly counting (figure 4C and D) (since at the time we decided to quantify virus DNA we did not have muscles from figure 3 C anymore). To our opinion, the explanation for a same effect on the phenotype but very different number of vg/nucleus may reside either in a non-homogeneous spreading of the AAV along muscle or in a threshold effect (i.e. once the efficient number of vg is reached to have the optimal effect a higher number of vg don't matter).

19. Potential silencing off-targets evaluation in MEFs or in vivo could be shown.

Answer: We have added as Appendix Figure S9 the result of nucleotide BLAST for the human and murine si9-RNA sequences.

20. Supplemental Figure 4: Why is the percent of histological abnormalities in HTZ control mice in the group 1M+1M higher than the HTZ control mice in the group 1M+3M? Isn't the accumulation of DPNH a marker for progress of the disease?

Answer: Yes the DPNH marker is supposed to be increased with the disease progression. We indeed counted more histological abnormalities in the group 1M+1M than in the second group 1M+3M. However, it would be necessary to performed DPNH staining at the same time using the same reagents to be able to compare the 2 groups (1M+1M and 1M+3M). It was unfortunately not the case. In contrast, all the staining for the different sub-groups at a given age were done together allowing the comparison. However, it might also be possible that for unknown reason the first group of mice 1M-1M was more severely affected by the disease than usual since we also observed greater force impairment.

21. Supplemental Figure 7: It is not clear why the mutant/WT ratio have been calculated using BglI digestion profile (and not PfoI as shown before). The size of the digested products should also be given.

Answer: BglI digestion profile has been replaced by PfoI digestion profile and the latter was used to calculate the mutant/WT ratio.

22. In panel C the density of non-transfected fibroblasts is higher than after transduction with si9 and scramble. Were the NT cells subjected to same procedure (transfectant w/o si9 or scramble)? Was the cell density after transduction with lower concentration of si9/scramble higher?

Answer: No, we didn't use transfectant in the NT cells. The cell densities after transduction with lower concentration (i.e. 50 and 80 nM) of si9/scramble was comparable to the ones observe at 100 nM. The pictures below show the cell density with 80 nM (up) and 100 nM (bottom) of si9-R465W RNAs.

Additional changes in the revised manuscript

The manuscript includes 3 new co-authors involved in the virus production (Laura Julien) and immortalization of mouse myoblast (Ayman Rabai and Jocelyn Laporte).

The Tables of p-Value and the Figures of uncropped gels have been updated according to the modifications.

Bibliography cited in the answer to reviewers

- Bitoun, Marc, Anne Cécile Durieux, Bernard Prudhon, Jorge A. Bevilacqua, Adrien Herledan, Vehary Sakanyan, Andoni Urtizberea, Luis Cartier, Norma B. Romero, and Pascale Guicheney. 2009. "Dynamin 2 Mutations Associated with Human Diseases Impair Clathrin-Mediated Receptor Endocytosis." *Human Mutation* 30 (10): 1419–27. doi:10.1002/humu.21086.
- Bitoun, Marc, Svetlana Maugenre, Pierre-Yves Jeannet, Emmanuelle Lacène, Xavier Ferrer, Pascal Laforêt, Jean-Jacques Martin, et al. 2005. "Mutations in Dynamin 2 Cause Dominant Centronuclear Myopathy." *Nature Genetics* 37 (11): 1207–9. doi:10.1038/ng1657.
- Carthew, Richard W., and Erik J. Sontheimer. 2009. "Origins and Mechanisms of miRNAs and siRNAs." *Cell*. doi:10.1016/j.cell.2009.01.035.
- Durieux, Anne-Cécile, Bernard Prudhon, Pascale Guicheney, and Marc Bitoun. 2010. "Dynamin 2 and Human Diseases." *Journal of Molecular Medicine (Berlin, Germany)* 88 (4): 339–50. doi:10.1007/s00109-009-0587-4.
- Durieux, Anne Cécile, Alban Vignaud, Bernard Prudhon, Mai Thao Viou, Maud Beuvin, Stéphane Vassilopoulos, Bodvaël Fraysse, et al. 2010. "A Centronuclear Myopathy-Dynamin 2 Mutation Impairs Skeletal Muscle Structure and Function in Mice." *Human Molecular Genetics* 19 (24): 4820–36. doi:10.1093/hmg/ddq413.
- Gibbs, Elizabeth M., Nigel F. Clarke, Kristy Rose, Emily C. Oates, Richard Webster, Eva L. Feldman, and James J. Dowling. 2013. "Neuromuscular Junction Abnormalities in DNM2-Related Centronuclear Myopathy." *Journal of Molecular Medicine* 91 (6): 727–37. doi:10.1007/s00109-013-0994-4.
- González-jamett, Arlek M, Ximena Baez-matus, María José Olivares, Maria José Guerra-fernández, Jacqueline Vasquez-navarrete, Mai Thao, Pascale Guicheney, et al. 2017. "Dynamin-2 Mutations Linked to Centronuclear Myopathy Impair Actin-Dependent Trafficking in Muscle Cells," no. May: 1–16. doi:10.1038/s41598-017-04418-w.
- Hu, Jiabin, Jing Liu, and David R. Corey. 2010. "Allele-Selective Inhibition of Huntingtin Expression by Switching to an miRNA-like RNAi Mechanism." *Chemistry and Biology* 17 (11): 1183–88. doi:10.1016/j.chembiol.2010.10.013.
- Jiang, Jianming, Hiroko Wakimoto, J. G. Seidman, and Christine E. Seidman. 2013. "Allele-Specific Silencing of Mutant *Myh6* Transcripts in Mice Suppresses Hypertrophic Cardiomyopathy." *Science* 342 (6154): 111–14. doi:10.1126/science.1236921.
- Jiang, Li, Houbin Zhang, Alexander M Dizhoor, Shannon E Boye, William W Hauswirth, Jeanne M Frederick, and Wolfgang Baehr. 2011. "Long-Term RNA Interference Gene Therapy in a Dominant Retinitis Pigmentosa Mouse Model." *Proceedings of the National Academy of Sciences of the United States of America* 108 (45): 18476–81. doi:10.1073/pnas.1112758108.
- Kenniston, Jon a, and Mark a Lemmon. 2010. "Dynamin GTPase Regulation Is Altered by PH Domain Mutations Found in Centronuclear Myopathy Patients." *The EMBO Journal* 29 (18). Nature Publishing Group: 3054–67. doi:10.1038/emboj.2010.187.
- Le Hir, Maëva, Aurélie Goyenvalle, Cécile Peccate, Guillaume Précigout, Kay E Davies, Thomas Voit, Luis Garcia, and Stéphanie Lorain. 2013. "AAV Genome Loss from Dystrophic Mouse Muscles during AAV-U7 snRNA-Mediated Exon-Skipping Therapy." *Molecular Therapy* 21 (8): 1551–58. doi:10.1038/mt.2013.121.
- Lorain, Stéphanie, David-Alexandre Gross, Aurélie Goyenvalle, Olivier Danos, Jean Davoust, and Luis Garcia. 2008. "Transient Immunomodulation Allows Repeated Injections of AAV1 and Correction of Muscular Dystrophy in Multiple Muscles." *Molecular Therapy* 16 (3): 541–47. doi:10.1038/sj.mt.6300377.
- Tasfaout, Hichem, Suzie Buono, Shuling Guo, Christine Kretz, Nadia Messaddeq, Sheri Booten, Sarah Greenlee, Brett P Monia, Belinda S Cowling, and Jocelyn Laporte. 2017. "Antisense Oligonucleotide-Mediated Dnm2 Knockdown Prevents and Reverts Myotubular Myopathy in Mice." *Nature Communications* 8. Nature Publishing Group: 15661. doi:10.1038/ncomms15661.
- Trochet, Delphine, Bernard Prudhon, Arnaud Jollet, Stéphanie Lorain, and Marc Bitoun. 2016.

“Reprogramming the Dynamin 2 mRNA by Spliceosome-Mediated RNA Trans-Splicing.”
Molecular Therapy - Nucleic Acids 5 (9): e362. doi:10.1038/mtna.2016.67.

Wang, Lei, Barbara Barylko, Christopher Byers, Justin a Ross, David M Jameson, and Joseph P Albanesi. 2010. “Dynamin 2 Mutants Linked to Centronuclear Myopathies Form Abnormally Stable Polymers.” *The Journal of Biological Chemistry* 285 (30): 22753–57. doi:10.1074/jbc.C110.130013.

Zincarelli, Carmela, Stephen Soltys, Giuseppe Rengo, and Joseph E Rabinowitz. 2008. “Analysis of AAV Serotypes 1-9 Mediated Gene Expression and Tropism in Mice after Systemic Injection.” *Molecular Therapy: The Journal of the American Society of Gene Therapy* 16 (6). The American Society of Gene Therapy: 1073–80. doi:10.1038/mt.2008.76.

2nd Editorial Decision

08 November 2017

Thank you for the submission of your revised manuscript to EMBO Molecular Medicine. We have now received the enclosed reports from the referees that were asked to re-assess it. As you will see the reviewers are now globally supportive and I am pleased to inform you that we will be able to accept your manuscript pending the following final amendments:

1) Please address referee 1 's comments and provide a letter INCLUDING the reviewer's reports and your detailed responses to their comments (as Word file).

***** Reviewer's comments *****

Referee #1 (Remarks for Author):

The revised paper is now improved anyway some points still need to be elucidated.

1. The authors did test siRNA in myoblasts from their mouse model as suggested. The results showed that siRNAs are still working but less efficient than in fibroblasts. Anyway this result is not discussed in the appropriate section. Also, why did they immortalise myoblasts instead of using primary ones? Authors should discuss these points in the results section.

2. Regarding Appendix Figure S5 on 1M-1M mice I still have some doubts about the results. As shown in Figure S5A and as said by the authors in the text, there is no detectable reduction in mutated Dnm2 after si9 and si10 especially for si10, which is consistent with no differences in Mutant/wt ratio and no differences in histological analysis (Fig 5E). I wonder how this no difference (or very minimal one) turned into a functional improvement and whether this improvement in muscle force is actually due to the action of si9 and si10 in 1M-1M mice.

5. Regarding Fig 2, authors should explain different analysis of Dnm2 mutated in the results and not in the materials and method otherwise the readers will not understand differences in the outcome of the two analyses.

Referee #2 (Remarks for Author):

I have read the point by point answers to my concerns and agreeable that manuscript is suitable for publication

Referee #3 (Remarks for Author):

This revised manuscript on allele-specific RNA interference for autosomal dominant centronuclear myopathy due to a DNMT2 mutation has been substantially improved. The authors added more data and address properly the concerns raised by the reviewers. This revised manuscript should be published.

Referee #1 (Remarks for Author):

The revised paper is now improved anyway some points still need to be elucidated.

1. The authors did test siRNA in myoblasts from their mouse model as suggested. The results showed that siRNAs are still working but less efficient than in fibroblasts. Anyway this result is not discussed in the appropriate section. Also, why did they immortalise myoblasts instead of using primary ones? Authors should discuss these points in the results section.

Answer:

We indeed observed a lower efficiency of the silencing in mouse myoblasts than in MEF at the same concentration (*i.e.* 100nM), however according to our experimental experience myoblast cells are slightly harder to transfect than fibroblasts.

Therefore, we chose to not discuss this point in the paper since we think it is likely due to differences in transfection efficiency between cells. A dose response study in myoblast cells is ongoing to determine optimal siRNA concentration to obtain maximum effect in these cells.

We chose to use immortalized myoblasts because they were already available and easier to manipulate than primary ones who rapidly stop dividing in classical culture conditions. Indeed, unlike primary cells, immortalized myoblasts allow the amplification of a large amount of cells. This is useful to repeat or plan further experiments in similar conditions with no need to isolate myoblast each time.

2. Regarding Appendix Figure S5 on 1M-1M mice I still have some doubts about the results. As shown in Figure S5A and as said by the authors in the text, there is no detectable reduction in mutated *Dnm2* after si9 and si10 especially for si10, which is consistent with no differences in Mutant/wt ratio and no differences in histological analysis (Fig 5E). I wonder how this no difference (or very minimal one) turned into a functional improvement and whether this improvement in muscle force is actually due to the action of si9 and si10 in 1M-1M mice.

Answer:

For 1M-1M mice, there is in fact a slight reduction in the mutant allele and the change in total *Dnm2* transcript (mutant+WT) was undetectable under these conditions. See in the result section page 8, lane 163:

...“Changes in the total *Dnm2* transcript levels were undetectable and the mutant transcript was only slightly reduced in treated mice (Appendix Figure S5 A and B).”

Regarding Figure S5A panel B the mutant reduction is significant relative to *Gapdh* while indeed not relative to the wild-type. This discrepancy may be due to sensitivity limits of our assay along with sample variability.

Moreover, force functional improvement was also observed for the 6M-3M mice with relatively low level of mutant reduction (~10% of mutant reduction).

3. Regarding Fig 2, authors should explain different analysis of *Dnm2* mutated in the results and not in the materials and method otherwise the readers will not understand differences in the outcome of the two analyses.

Answer:

We clarified the different analysis of *Dnm2* transcript in the result section as follows:

Page 6: Lanes 102-105:

“We developed RT-PCR assay to discriminate the WT and mutant alleles after restriction enzyme digestion performed at the end of the exponential phase of PCR amplification (Appendix Figure S1). Using this assay, we assessed allele-specific...”

lane 111-112:

“After 48 h semi-quantitative RT-PCR showed around 50% reduction of total *Dnm2*-mRNA expression (WT + mutant) for each siRNA (Figure 2A).”

Referee #2 (Remarks for Author):

I have read the point by point answers to my concerns and agreeable that manuscript is suitable for publication

Referee #3 (Remarks for Author):

This revised manuscript on allele-specific RNA interference for autosomal dominant centronuclear myopathy due to a DNM2 mutation has been substantially improved. The authors added more data and address properly the concerns raised by the reviewers. This revised manuscript should be published.

Corresponding Author Name: Marc Bitoun
Journal Submitted to: EMBO Molecular Medicine
Manuscript Number: EMM-2017-07988